# CHRONOPLAY: A FRAMEWORK FOR MODELING DUAL DYNAMICS AND AUTHENTICITY IN GAME RAG BENCHMARKS

**Liyang He**[*1], **Yuren Zhang**[*1], **Ziwei Zhu**[2], **Zhenghui Li**[3], **Shiwei Tong**[†1]

[1]Tencent,    [2]The Chinese University of Hong Kong,    [3]Independent Researcher

{liyanghe, yurenzzhang}@tencent.com,    221049028@link.cuhk.edu.cn,
417341879@qq.com,    shiweitong@tencent.com

## ABSTRACT

Retrieval Augmented Generation (RAG) systems are increasingly vital in dynamic domains like online gaming, yet the lack of a dedicated benchmark has impeded standardized evaluation in this area. The core difficulty lies in *Dual Dynamics*: the constant interplay between game content updates and the shifting focus of the player community. Furthermore, the necessity of automating such a benchmark introduces a critical requirement for player-centric authenticity to ensure generated questions are realistic. To address this integrated challenge, we introduce ChronoPlay, a novel framework for the automated and continuous generation of game RAG benchmarks. ChronoPlay utilizes a dual-dynamic update mechanism to track both forms of change, and a dual-source synthesis engine that draws from official sources and player community to ensure both factual correctness and authentic query patterns. We instantiate our framework on three distinct games to create the first dynamic RAG benchmark for the gaming domain, offering new insights into model performance under these complex and realistic conditions. Our code is available at: https://github.com/hly1998/ChronoPlay.

## 1 INTRODUCTION

The advancement of Retrieval Augmented Generation (RAG) has been largely driven by its benchmarks (Yu et al., 2024; Gan et al., 2025). From foundational benchmarks (Yang et al., 2018; Kwiatkowski et al., 2019) and specialized domain evaluations (Li et al., 2025; Wang et al., 2024; Zhong et al., 2025; Jeon et al., 2025) to sophisticated diagnostic frameworks (Ru et al., 2024), these resources provide the essential, standardized means to measure progress. Recent efforts have further pushed the frontier from static snapshots to dynamic benchmarks that evolve over time (Ouyang et al., 2025; Ko et al., 2024), reflecting the currency of information in the real world. The global gaming industry, a vast and highly dynamic digital frontier, represents a critical domain for such advancements (Goh et al., 2023). Within this ecosystem, RAG is emerging as a key technology to enhance player experiences, from intelligent assistants to automated support bots (Liu et al., 2025; Feng et al., 2025). However, a significant gap emerges: there are currently no RAG benchmarks for the gaming domain, leaving the application of RAG systems in this area without standardized evaluation.

This absence stems from the unique nature of the gaming ecosystem, which is composed of two constantly evolving entities: the game itself and its player community. This structure gives rise to a core challenge we term *Dual Dynamics*, as illustrated in Figure 1. On one hand, *Knowledge Evolution* occurs as the game's content and rules are in a constant state of flux due to frequent patches and version updates Wang et al. (2023). This is different from some stable domains, where a comprehensive static benchmark is enough to evaluate a general-purpose model. In the world of live-service games, such a static evaluation would become obsolete. On the other hand, a benchmark must also track *User Interest Drift*, which represents the systematic evolution of the player community's

---

[*]Equal contribution.

[†]Corresponding author.

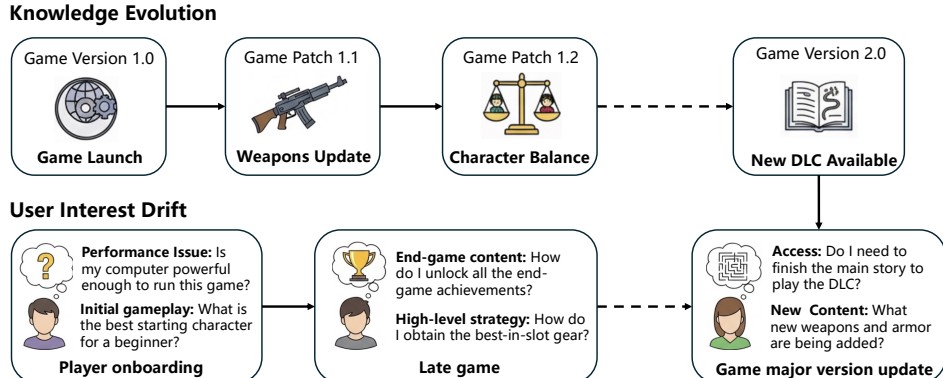

Figure 1: An illustration of *Dual Dynamics* in Game, including *Knowledge Evolution*, which traces the game's knowledge updates, and *User Interest Drift*, which maps the changing interests of players.

focus, from initial onboarding questions to late game content Kummer et al. (2017); Yılmaz (2025). Capturing this shifting focus is crucial, as a benchmark that fails to do so would lead to models being optimized on a distribution of problems that no longer reflects what the community cares about, resulting in a biased and irrelevant evaluation.

Given the sheer velocity of these dual dynamics, manually curating a benchmark that remains consistently up-to-date is practically impossible. This challenge is compounded by the vast diversity across different games. Consequently, automated synthesis has emerged as the only viable path forward for creating dynamic benchmarks in this domain. However, existing synthesis approaches Kasai et al. (2023); Ko et al. (2024); Kim et al. (2024); Ouyang et al. (2025); Shabtay et al. (2025) have focused primarily on a single dimension of change: the evolution of knowledge. By concentrating only on tracking knowledge updates, they have overlooked the critical need for authenticity in their generation process. In a user-centered domain, a benchmark is fundamentally invalid if it is filled with unrealistic questions that no real player would ask, regardless of their grammatical correctness.

To tackle the challenges mentioned above, we introduce **ChronoPlay**, a novel framework for modeling both dual dynamics and authenticity in the generation of game RAG benchmarks. Its core features include: (1) a dual-source synthesis engine that ensures factual correctness by drawing from official sources, while capturing authentic player question patterns and interest preferences by mining player community information; and (2) a dual-dynamic update mechanism that precisely refreshes question-answer pairs by identifying entities in game updates and dynamically adjusts the distribution of evaluation questions in response to shifts in community interest.

Using this framework, we instantiate three distinct games. Our subsequent analysis provides an initial baseline for classical RAG systems, and the results demonstrate that our benchmark effectively captures the unique challenges of these dually dynamic conditions, highlighting aspects of model performance that existing evaluation methods cannot measure. Notably, while we focus on the gaming domain, our methodology is applicable to other domains characterized by an evolving knowledge base and an active user community, such as e-commerce and social media platforms. Our contributions are as follows:

1. We identify dual dynamics as a fundamental challenge for RAG systems operating in dynamic domains such as gaming, and argue that addressing it requires an automated synthesis process with player-centric authenticity as a critical requirement.

2. We propose ChronoPlay, the first framework designed to automatically generate a dynamic benchmark that integrally addresses the challenges of temporal relevance and player-centric authenticity.

3. We instantiate our framework on three distinct games to create the first dynamic RAG benchmarks for the gaming domain. Our analysis offers new insights into the performance of existing RAG systems under these complex, realistic conditions.

## 2 RELATED WORK

The development of RAG systems benefited significantly from question-answering benchmarks like Natural Questions (NQ) (Kwiatkowski et al., 2019), HotpotQA (Yang et al., 2018), PopQA (Mallen et al., 2023), and CRAG (Yang et al., 2024). They provided a standardized environment for evaluating a model's retrieval and reasoning capabilities within a closed, static knowledge world. However, their static nature makes them incapable of assessing a system's adaptability to the continuously evolving knowledge of the real world.

To address this limitation, some works have focused on creating dynamic benchmarks (Shirali et al., 2022). These efforts can be broadly categorized by their update trigger: being either period-driven or fact-driven. The period-driven approach updates at fixed intervals, often leveraging retrospective snapshots of sources like Wikipedia. For instance, HOH (Ouyang et al., 2025), GrowOVER (Ko et al., 2024), EvolvingQA (Kim et al., 2024), and DynaQuest (Lin et al., 2025a) use monthly snapshots to generate question-answer pairs that test a model's ability to handle outdated information. Besides, DRAGON (Chernogorskii et al., 2025) is built upon a periodically updated news corpus. The fact-driven approach aims for lower latency by triggering updates upon the emergence of new information. While not all are designed strictly for RAG, methods like REALTIMEQA (Kasai et al., 2023), which uses live news feeds, and LIVEXIV (Shabtay et al., 2025), which scrapes new preprints, exemplify this paradigm.

However, this exclusive focus on knowledge evolution overlooks the user interest drift. The dynamic evolution of user interests is a significant phenomenon in many fields like social computing and recommendation systems (Singer et al., 2014; Yin et al., 2016; Sun & Dong, 2017; Ding et al., 2022; Lin et al., 2025b). This challenge is also mirrored in granular tasks such as aspect-level sentiment analysis (Zhang et al., 2021). This is particularly pronounced in gaming, a trend confirmed by our preliminary analysis. As shown in Figure 2, discussion hotspots in the *Dying Light 2* player community clearly drifted from early topics like *System Requirements* to later concerns about *Gameplay Mechanics*. By focusing exclusively on knowledge updates, all current temporally-aware benchmarks ignore the evolution of user interest, which is critical for building applications that serve genuine user needs.

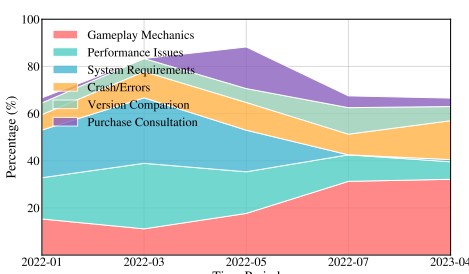

Figure 2: Distribution of main topics over time in Dying Light 2.

## 3 THE CHRONOPLAY FRAMEWORK

The ChronoPlay framework is designed to transform the challenge of dual dynamics and authenticity in the gaming domain into an executable and automated solution. Its core objective is to continuously generate a dynamically evolving benchmark, denoted as $\mathcal{B} = \{\mathcal{B}_1, \mathcal{B}_2, ..., \mathcal{B}_t, ...\}$. Each benchmark slice $\mathcal{B}_t$ at a specific time point $t$ is a pair $(\mathcal{K}_t, \mathcal{D}_t)$, which consists of a corpus knowledge base $\mathcal{K}_t$ for retrieval and a corresponding evaluation dataset $\mathcal{D}_t$. This dataset is composed of a series of evaluation tuples, with each tuple defined as $d = (\mathcal{Q}, \mathcal{A}, \mathcal{C}_{ref}, \theta, \tau, \sigma)$. These components represent the question $\mathcal{Q}$, answer $\mathcal{A}$, reference knowledge snippets $\mathcal{C}_{ref} \in \mathcal{K}_t$, question topic $\theta$, timestamp $\tau$, and associated in-game entities $\sigma$, respectively.

To achieve this, we have designed two closely integrated core components: a data synthesis pipeline that builds upon dual-source data assets, and a dual-dynamic update mechanism responsible for the continuous evolution of the benchmark. We describe the overall framework and pipeline in the main text, while the technical details are provided in Appendix D.

### 3.1 DUAL-SOURCE DATA SYNTHESIS FRAMEWORK

To ensure the benchmark is both factually accurate and representative of authentic player questions, we have designed a dual-source data synthesis framework. Instead of relying on noisy questions

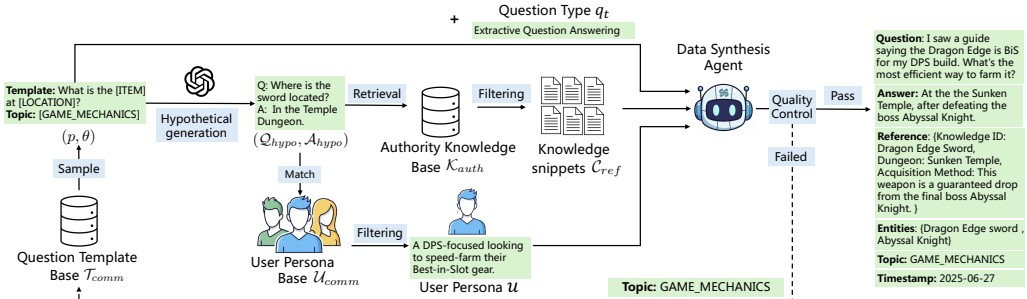

Figure 3: An illustration of the Dual-Source Synthesis Pipeline. It leverages the Authority Knowledge Base $\mathcal{K}_{auth}$ for factual grounding, and uses the Question Template Base $\mathcal{T}_{comm}$ and the User Persona $\mathcal{U}_{comm}$ for authentic question patterns.

from online communities, our framework employs a multi-stage pipeline that integrates authoritative knowledge with player community information.

### 3.1.1 AUTHORITY KNOWLEDGE BASE ($\mathcal{K}_{auth}$)

To guarantee factual accuracy, we construct an *Authority Knowledge Base* $\mathcal{K}_{auth}$. Each knowledge snippet $k_i \in \mathcal{K}_{auth}$ is formalized as a tuple $(k_c, k_\tau, k_\sigma)$, representing the content, timestamp, and associated entities. We systematically aggregate game wikis and official patch notes, processing raw HTML pages and tabular data into uniform, retrievable knowledge snippets $k_c$ using DOM tree analysis and LLM for formatting. For official data, we precisely extract the publication time as the timestamp $k_\tau$. Based on the Self-ICL approach Chen et al. (2023), we use an NER function $\mathcal{E}(\cdot)$ on all knowledge snippets to extract in-game entities $k_\sigma = \mathcal{E}(k_c)$. These entities are crucial for accurately identifying and updating affected knowledge.

### 3.1.2 QUESTION TEMPLATE BASE ($\mathcal{T}_{comm}$) & USER PERSONA BASE ($\mathcal{U}_{comm}$)

To capture the authenticity of user queries, we build two community bases by collecting real questions from player communities. We invite domain experts to develop a hierarchical topic taxonomy $\Theta$, based on a large sample of player questions. This taxonomy comprises 6 main categories and 21 sub-categories, covering aspects from technical issues to game content and purchase consultation. Appendix B details the taxonomy. Then, we prompt an LLM to mine these question posts, decoupling and extracting two reusable elements: question templates $p$ with their corresponding topics $\theta \in \Theta$, and user personas $u$. This decoupling allows these authentic, game-agnostic query patterns to be reused across different games, which greatly enhances the framework's scalability. The resulting template-topic pairs $(p, \theta)$ form the *Question Template Base* $\mathcal{T}_{comm}$, and the user personas $u$ constitute the *User Persona Base* $\mathcal{U}_{comm}$. We also employ a vector-based filtering mechanism to deduplicate semantically similar items in both bases, followed by a final review by experts.

### 3.2 MULTI-STAGE SYNTHESIS PIPELINE

Based on the constructed bases, our synthesis pipeline organically combines these two data sources through a sophisticated multi-stage process as shown in Figure 3. We first vectorize the snippets from the knowledge base $\mathcal{K}_{auth}$ and build an index. To effectively bridge user intent with factual evidence, we draw inspiration from HyDE (Gao et al., 2023) and use a sampled template-topic pair $(p, \theta)$ from $\mathcal{T}_{comm}$ to prompt an LLM to generate a hypothetical question-answer pair $(\mathcal{Q}_{hypo}, \mathcal{A}_{hypo})$. Although the content of this pair is fictional, its embedding provides a more precise vector to locate relevant knowledge snippets $\mathcal{C}_{ref} = \{k_1, k_2, ..., k_n\}$ within the vector space of $\mathcal{K}_{auth}$.

Since players often ask context-dependent questions (e.g., "As a new player, which weapon is easiest to get?"), we incorporate a user persona $u \in \mathcal{U}_{comm}$ into the generation process. We embed the user persona base $\mathcal{U}_{comm}$ and build a vector index. The hypothetical pair $(\mathcal{Q}_{hypo}, \mathcal{A}_{hypo})$ is then used to query this index to find the most suitable persona. Only personas whose similarity score surpasses a

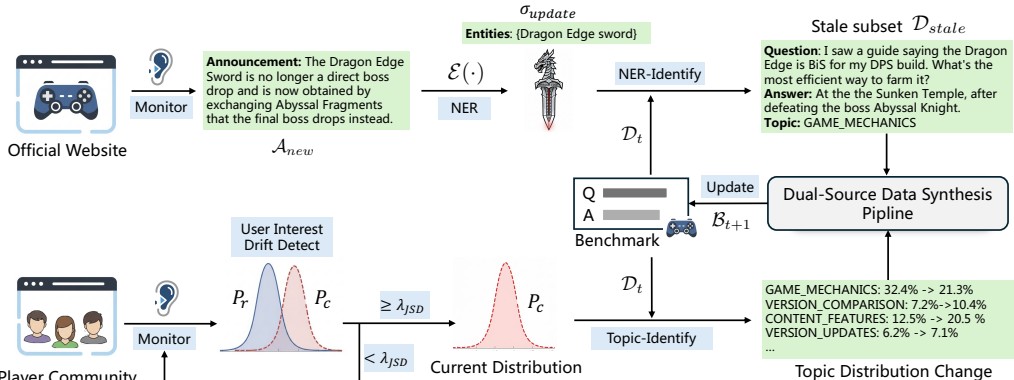

Figure 4: An illustration of the Dual-Dynamic Update Mechanism, showing both Knowledge Evolution and User Interest Drift pathways.

threshold $\lambda_p$ are considered candidates, and we select the top-ranked result. If no persona meets the threshold, this generation step proceeds without a user persona.

The final synthesis stage is orchestrated by a specialized data synthesis agent, which is designed to autonomously produce high-quality data tuples through an iterative refinement process. For each generation task, the agent begins by sampling a question type $q_t$ (e.g., extractive, comparative) from a predefined set (Wang et al., 2024). It then synthesizes a candidate tuple $d = (\mathcal{Q}, \mathcal{A}, \mathcal{C}_{ref}, \theta, \tau, \sigma)$ by conditioning a generator model on the comprehensive context, which includes the template-topic pair $(p, \theta)$, the retrieved factual snippets $\mathcal{C}_{ref}$, the user persona $u$, and the question type $q_t$. The in-game entities $\sigma$ and the timestamp $\tau$ are derived from the reference context $\mathcal{C}_{ref}$, with $\tau$ being defined as the most recent timestamp available, i.e., $\tau = \max(\{k_\tau | k \in \mathcal{C}_{ref}\})$. If no snippet in $\mathcal{C}_{ref}$ contains a timestamp, $\tau$ is left undefined.

Another core component of this agent is its integrated quality control mechanism. After generating a candidate tuple, the agent immediately assesses it against multiple quality dimensions. This assessment is performed by an LLM, following the increasingly common LLM-as-Judge paradigm for scalable and automated evaluation (Zheng et al., 2023; Dubois et al., 2024). If the tuple fails to meet a predefined quality threshold, it is discarded. Instead of terminating, the agent initiates a self-correction loop: it re-samples a new question template corresponding to the same topic $\theta$ from the question template base $\mathcal{T}_{comm}$ and re-attempts the synthesis. This iterative process continues until a high-quality tuple is successfully generated and validated.

## 3.3 DUAL-DYNAMIC UPDATE MECHANISM

Another core innovation of ChronoPlay is its ability to evolve over time, ensuring the long-term validity of the benchmark. This dynamism is driven by a dual update mechanism that responds to changes in both the game's knowledge and the player's interests as shown in Figure 4.

The first mechanism is designed to respond to *Knowledge Evolution*, guaranteeing the benchmark up-to-date and factually correct with respect to the game content. Game knowledge updates are typically driven by discrete events like official announcements. Our monitoring module detects such events and triggers an update workflow. Let $\mathcal{A}_{new}$ be a new announcement at time $t$. We first use the NER function $\mathcal{E}(\cdot)$ to identify the affected in-game entities $\sigma_{update} = \mathcal{E}(\mathcal{A}_{new})$. We then identify the stale subset of the dataset $\mathcal{D}_{stale}$, where each tuple's entity set intersects with the updated entities:

$$\mathcal{D}_{stale} = \{d \in \mathcal{D}_t \mid \sigma(d) \cap \sigma_{update} \neq \emptyset\} \tag{1}$$

The remaining valid tuples are denoted as $\mathcal{D}_{valid} = \mathcal{D}_t \setminus \mathcal{D}_{stale}$. The topics from these stale tuples are used to re-initiate the synthesis pipeline described in Section 3.2. This generates a new set of tuples $\mathcal{D}_{new}$, which reflects the latest information. The dataset is then updated to its next state: $\mathcal{D}_{t+1} = \mathcal{D}_{valid} \cup \mathcal{D}_{new}$. Besides, the corpus knowledge base is updated by $\mathcal{K}_{t+1} = \mathcal{K}_t \cup \mathcal{A}_{new}$. Finally, we obtain the new benchmark $\mathcal{B}_{t+1}$

The second mechanism is designed to respond to *User Interest Drift*, ensuring the benchmark's topical relevance. We continuously monitor the topic distribution $P_{[a,b]}(\Theta)$ of questions from the community within a sliding time window of size $W$, where $a$ and $b$ represent the start and end timestamps. To quantitatively detect significant shifts, we compare the topic distribution of the current window $P_c \equiv P_{[max(p_\tau, c_\tau - W), c_\tau]}(\Theta)$, with that of a reference period $P_r \equiv P_{[p_\tau, c_\tau]}(\Theta)$, where $p_\tau$ denotes the start time of the previous reference time segment and $c_\tau$ denotes the current time. A collective interest drift is flagged if a topic-weighted Jensen-Shannon Divergence between these distributions exceeds a predefined threshold $\lambda_{JSD}$. In this variant, the standard JSD calculation is modified: each topic $\theta$ is assigned a weight $w_\theta$ based on its overall prominence. Assume $M = \frac{1}{2}(P_c + P_r)$ is the mixture distribution in JSD, then the weight $w_\theta$ for each topic is calculated as $w_\theta = M(\theta)^\gamma / \sum_{\theta' \in \Theta} M(\theta')^\gamma$, with $\gamma$ being a hyperparameter. This method focuses the drift detection on significant trends, making it more robust against noise from low-frequency topics.

Upon detecting a drift, the framework initiates a benchmark resampling process to generate $\mathcal{B}_{t+1}$. The goal is to align the benchmark's topic distribution with the current topic distribution $P_c$. This involves down-sampling data from topics with waning interest and synthesizing new data for topics with emerging interest until the overall distribution of $\mathcal{B}_{t+1}$ reflects that of the active player community.

## 4 EXPERIMENTS

We conduct experiments to answer four core research questions: **RQ1**: How do RAG systems perform on our dynamic benchmark, and how does their performance fluctuate across a game's lifecycle? **RQ2**: How do knowledge evolution and user interest drift individually and collectively impact these performance fluctuations? **RQ3**: How do our key synthesis modules contribute to the authenticity and quality of the generated benchmark data? **RQ4**: How efficient is our dual-dynamic update mechanism, and what are the respective roles of knowledge and interest as drivers of change?

### 4.1 EXPERIMENTAL SETUP

#### 4.1.1 DATASET INSTANTIATION

We instantiate our benchmark on three distinct games: *Dying Light 2*, *Dune: Awakening*, and *PUBG Mobile*, which will be abbreviated as *DL2*, *Dune*, and *PUBG* in the following content. These games represent different characteristics. For each game, we collect data from official sources and major player communities. Table 1 summarizes the key statistics for each dataset. For our analysis, we partition each game's timeline into discrete phases $\{B_1, ..., B_n\}$ based on significant shifts in user interest. Shifts in user interest directly measure what players deem relevant and often represent the consolidated impact of multiple underlying knowledge updates, thus forming coherent periods for analysis. For more details on the game data, hyperparameter settings, and the specific partitioning of phases, please refer to Appendix A.

Table 1: Key statistics of the three instantiated game benchmarks.

| Game | Time Range | # Comm. Posts | Wiki Tokens | Update Tokens | # Synth. Qs | # Phases |
|------|-----------|--------------|-------------|--------------|-------------|----------|
| Dying Light 2 | Jan 22 - Jul 25 | 10,478 | 369,120 | 297,371 | 2,000 | 5 |
| Dune: Awakening | Jun 25 - Aug 25 | 37,079 | 18,123,190 | 53,833 | 3,000 | 6 |
| PUBG Mobile | Jan 24 - Jul 25 | 60,632 | 86,652 | 56,703 | 1,400 | 7 |

#### 4.1.2 BASELINES AND METRICS

We construct RAG systems using four retrievers, including BM25 (Robertson et al., 1995), BGE-M3 (Chen et al., 2024), Qwen3-Embedding (Qwen3-Embedding-0.6B) (Zhang et al., 2025), and text-embedding-3 (text-embedding-3-samll). We used six generators, including closed-source and open-source large language models: GPT-4o (GPT-4o-2024-11-20) (Hurst et al., 2024), Qwen3 (Qwen3-14B) (Yang et al., 2025), llama-4 (llama-4-scout-17b) Meta (2025), gemini-2.5-flash Comanici et al. (2025), Claude-3.5-Sonnet (claude-3-5-sonnet-20240620) Anthropic (2024), and DeepSeek-V3 Liu et al. (2024). For retrieval, we evaluate using Recall@K, F1@K, and NDCG@K. For generation, we use an LLM-as-Judge to assess correctness and faithfulness, with details provided in Appendix C.

## 4.2 RQ1: LIFECYCLE PERFORMANCE EVALUATION

We first evaluate how RAG systems perform on each sequential benchmark slice for each game. This phase-by-phase analysis is designed to reveal how the performance of different RAG systems fluctuates as the game evolves. We present the retrieval and generation results separately.

### 4.2.1 RETRIEVAL PERFORMANCE

Table 2 shows the results when the number of retrieved documents $K$ is set to 3. Appendix E shows the results when $K$ is set to 1 and 5. From these results, we can draw several key conclusions. First, no single retriever is the best for all situations. For example, text-embedding-3 is the top-performing retriever in most phases, while Qwen3-Embedding achieves a higher score in several phases of *PUBG*. Second, the performance of all retrievers changes significantly across different phases, which confirms our main motivation. For example, in *DL2*, all models show a clear performance drop in Phase 4. The topic distribution analysis reveals that questions about *GAMEPLAY_MECHANICS* surged from 17.64% in Phase 3 to 31.25% in Phase 4. These questions are often more complex, making it harder for retrievers to find the precise documents needed. Third, we found that BGE-M3's performance on *Dune* was very low compared to the other models. This may be because *Dune* was recently released in June 2025, and it includes more proper nouns (e.g., terrarium of muad'dib), which severely tests the generalization ability of the models.

Table 2: Retrieval performance on three games across their phases. We report Recall@3 (R@3), F1@3, and NDCG@3 (N@3). The best performing result in each row is in **bold**.

| Phase | BM25 | | | Qwen3-Embedding | | | BGE-M3 | | | text-embedding-3 | | |
|---|---|---|---|---|---|---|---|---|---|---|---|---|
| | R@3 | F1@3 | N@3 | R@3 | F1@3 | N@3 | R@3 | F1@3 | N@3 | R@3 | F1@3 | N@3 |
| *DL2* | | | | | | | | | | | | |
| 1 | 0.389 | 0.363 | 0.498 | 0.342 | 0.320 | 0.454 | 0.378 | 0.355 | 0.507 | **0.521** | **0.495** | **0.698** |
| 2 | 0.387 | 0.356 | 0.509 | 0.291 | 0.262 | 0.372 | 0.339 | 0.311 | 0.437 | **0.536** | **0.505** | **0.683** |
| 3 | 0.403 | 0.371 | 0.526 | 0.338 | 0.308 | 0.449 | 0.367 | 0.338 | 0.497 | **0.551** | **0.519** | **0.716** |
| 4 | 0.323 | 0.305 | 0.428 | 0.273 | 0.256 | 0.398 | 0.309 | 0.290 | 0.429 | **0.419** | **0.396** | **0.604** |
| 5 | 0.300 | 0.283 | 0.388 | 0.262 | 0.245 | 0.367 | 0.278 | 0.260 | 0.385 | **0.422** | **0.401** | **0.593** |
| *Dune* | | | | | | | | | | | | |
| 1 | 0.282 | 0.268 | 0.386 | 0.339 | 0.325 | 0.612 | 0.067 | 0.065 | 0.056 | **0.368** | **0.354** | **0.645** |
| 2 | 0.271 | 0.256 | 0.370 | 0.319 | 0.302 | 0.570 | 0.028 | 0.028 | 0.027 | **0.348** | **0.331** | **0.635** |
| 3 | 0.314 | 0.296 | 0.446 | 0.341 | 0.321 | 0.678 | 0.038 | 0.038 | 0.036 | **0.381** | **0.360** | **0.738** |
| 4 | 0.270 | 0.255 | 0.362 | 0.314 | 0.298 | 0.563 | 0.040 | 0.040 | 0.038 | **0.346** | **0.329** | **0.612** |
| 5 | 0.268 | 0.249 | 0.355 | 0.320 | 0.300 | 0.580 | 0.039 | 0.038 | 0.038 | **0.343** | **0.325** | **0.622** |
| 6 | 0.265 | 0.252 | 0.355 | 0.313 | 0.300 | 0.549 | 0.050 | 0.049 | 0.046 | **0.346** | **0.334** | **0.604** |
| *PUBG* | | | | | | | | | | | | |
| 1 | 0.415 | 0.415 | 0.503 | 0.515 | 0.515 | 0.580 | 0.482 | 0.482 | 0.570 | **0.528** | **0.528** | **0.590** |
| 2 | 0.262 | 0.262 | 0.307 | **0.372** | **0.372** | **0.402** | 0.342 | 0.342 | 0.379 | 0.365 | 0.365 | 0.395 |
| 3 | 0.533 | 0.533 | 0.576 | **0.578** | **0.578** | **0.623** | 0.557 | 0.557 | 0.606 | 0.577 | 0.577 | 0.615 |
| 4 | 0.237 | 0.237 | 0.280 | 0.332 | 0.332 | 0.368 | 0.308 | 0.308 | 0.352 | **0.338** | **0.338** | **0.379** |
| 5 | 0.338 | 0.338 | 0.421 | **0.392** | **0.392** | **0.460** | 0.380 | 0.380 | 0.450 | 0.377 | 0.377 | 0.432 |
| 6 | 0.352 | 0.352 | 0.426 | 0.480 | 0.480 | 0.536 | 0.440 | 0.440 | 0.505 | **0.498** | **0.498** | **0.549** |
| 7 | 0.322 | 0.322 | 0.385 | **0.458** | **0.458** | **0.517** | 0.427 | 0.427 | 0.493 | 0.442 | 0.442 | 0.500 |

### 4.2.2 GENERATION PERFORMANCE

We use the top 3 retrieved documents from the top-performing retriever, text-embedding-3, as the context for the six generator models. Figure 5 shows the most critical metric: correctness scores, while the faithfulness scores are discussed in Appendix F.

From the results, we can draw several conclusions. First, a key observation is that generator performance is not static, similar to the retrieval results. In the game *PUBG*, the fluctuations in each phase are particularly significant. This is because, as a highly interactive live-service game, player interest is more prone to shifting. Second, as a general trend, the generator performance is consistent with the retrieval results. However, sometimes the correctness score can drop even with good retrieval. A clear example is in *PUBG* when comparing Phase 1 and Phase 3. Although the retriever performed slightly better in Phase 3 (Table 2), the correctness scores for all generators are significantly lower.

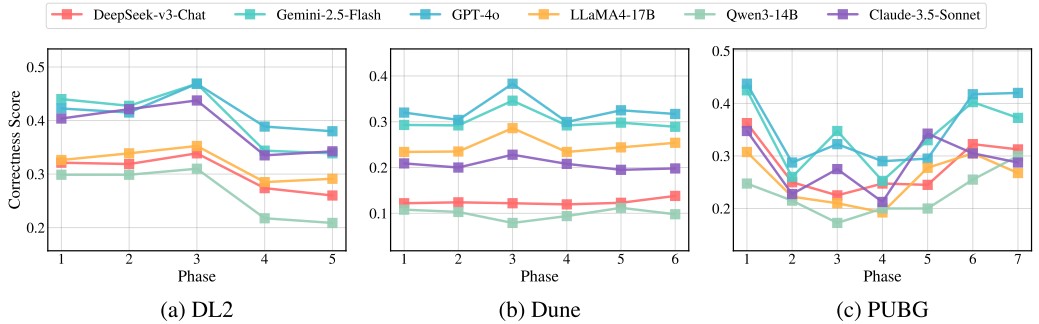

Figure 5: Generator correctness scores across the phases of each game.

This implies that the questions in Phase 3, while answerable with the provided documents, require more complex reasoning, which challenges the generators.

Overall, these retrieval and generation results confirm that a dynamic, phase-based evaluation is essential for the gaming domain. A static benchmark would average out these critical performance differences, failing to identify the specific types of knowledge updates, user interests, and question complexities that challenge modern RAG systems.

## 4.3 RQ2: Deconstructing the Impact of Dual Dynamics

To understand the cause of the fluctuations observed in RQ1, we isolate the individual effects of knowledge and interest updates using the relatively more volatile *PUBG* benchmark. We test a RAG system (using text-embedding-3 and GPT-4o) on our **Dual-Dynamic** benchmark against two variants: one that only tracks knowledge updates (**Knowledge-Only** benchmark) and one that only tracks user interest (**Interest-Only** benchmark).

Figure 6 shows the results. The line chart illustrates the results for the three benchmarks, and the violin plot represents the statistical volatility of these benchmarks. The key finding is that both knowledge updates and user interest drift are major factors affecting volatility. However, compared to our full Dual-Dynamic Benchmark, only considering one type of update leads to some evaluation bias. For example, the Knowledge-Only benchmark, which ignores user interest, hides the performance changes in Phase 4 and Phase 7, as these two phases had no knowledge updates. The Interest-Only benchmark, on the

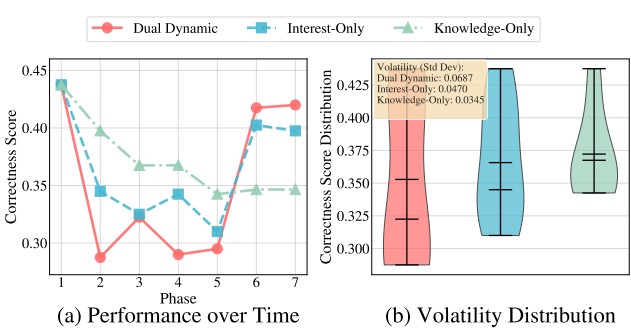

Figure 6: Comparison of RAG performance on three benchmark variants for *PUBG Mobile*.

other hand, ignores changes caused by game knowledge updates. Therefore, a benchmark must track both types of changes.

## 4.4 RQ3: Ablation Study on Synthesis Modules

Having demonstrated the importance of our benchmark, we now validate its construction with an ablation study. We compare our **Full Pipeline** against three degraded versions: (1) **w/o Hypothetical Q&A**, (2) **w/o User Persona**, and (3) **w/o Question Template**. Since these synthesis modules were specifically designed to make our benchmark player-centric, our primary evaluation criterion is authenticity, which means how much a question sounds like it was written by a real player. The evaluation uses a competitive method where both LLM-as-judges (we use GPT-4o, Gemini 2.5-Pro,

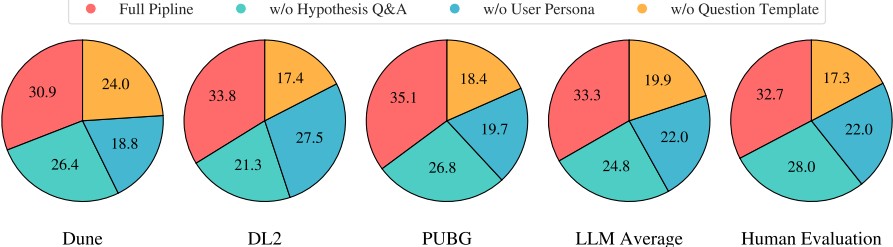

Figure 7: Ablation study results for the authenticity criterion. The pie charts show the win rates of our four synthesis settings across the three games, the average score, and the human expert evaluation.

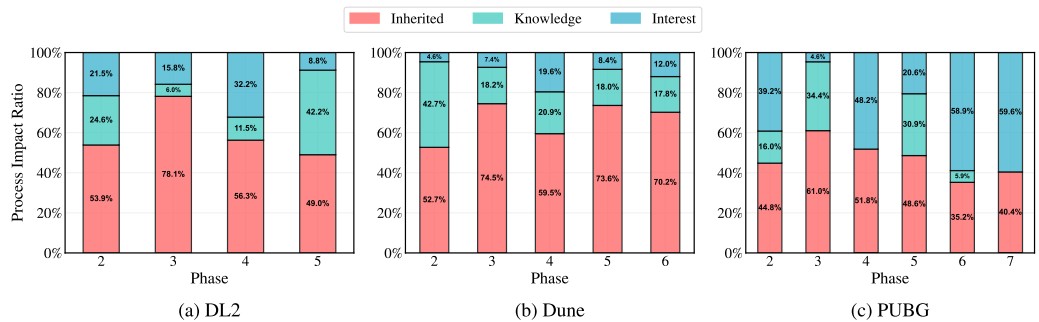

Figure 8: Analysis of the dual-dynamic update process. The composition of each new benchmark phase is broken down by its origin: questions that were inherited, updated due to knowledge changes, or updated due to interest shifts.

and DeepSeek-R1 as evaluation model) and human experts select the single best question from the four settings. In addition, we conducted a secondary analysis on question clarity. Appendix H details the experimental setup and the clarity results. Figure 7 shows the results for our primary authenticity evaluation. The results are clear and consistent: both the LLM judges and human experts show a strong preference for the Full Pipeline. The w/o Question Template setting performs the worst by a large margin. This result is expected, as the community-mined templates are the primary source of realistic user phrasing and intent. Without them, the generated questions become generic and lose their authentic feel. This finding validates that our full synthesis pipeline is crucial for creating a benchmark that faithfully represents real user questions.

### 4.5 RQ4: ANALYSIS OF THE DUAL-DYNAMIC UPDATE PROCESS

Finally, we analyze the efficiency and properties of our update mechanism. We categorize questions in each new phase as either **Inherited**, updated due to **Knowledge**, or updated due to **Interest**. Figure 8 visualizes this breakdown. The results show two key findings. First, the process is highly efficient. Across most phases, a large portion of the benchmark is inherited, meaning a full data reconstruction and evaluation is unnecessary. Furthermore, the primary driver of change varies dramatically. For example, the update to Phase 3 of *PUBG* was largely knowledge-driven (34.4%), while the update to Phase 4 was entirely interest-driven (48.2%). This confirms that both dynamics are independent and crucial forces that our mechanism successfully captures, ensuring the benchmark remains faithful to the true state of the player-centric game environment.

### 5 CONCLUSION AND FUTURE WORK

In this paper, we address the critical challenge of evaluating RAG systems in dynamic domains like gaming. We identify the core problem as *Dual Dynamics*: the constant, co-evolving interplay of factual game knowledge and the shifting interests of the player community. To solve this, we proposed

*ChronoPlay*, the first framework to automatically generate dynamic and authentic benchmarks that model both of these forces. Our experiments across three games demonstrate that RAG system performance is highly volatile over a game's lifecycle. We prove that this volatility is a product of both knowledge evolution and user interest drift, and benchmarks that ignore either dimension provide a misleadingly stable and unrealistic evaluation. An important future frontier is extending *ChronoPlay* to personalized user histories, evolving the persona base into a dynamic user state module. This paves the way for stateful RAG architectures that leverage persistent profiles and update mechanisms for personalized retrieval and generation. Ultimately, *ChronoPlay* offers not just a dynamic benchmark for gaming but a new paradigm for creating more faithful evaluations in any evolving, user-centric environment, serving as a foundation for more adaptive and personalized RAG systems.

## ETHICS STATEMENT

We confirm that our work adheres to the ICLR Code of Ethics. All data used in this research, including game information and community posts, were collected from publicly accessible online sources such as Fandom wikis, official game websites, and public forums. No private user data was accessed or used in this study. The human annotation and evaluation tasks described in this paper were performed by the authors of this work, who consented to their voluntary participation. The User Persona Base generated by our framework is a synthetic description of player types, inferred from public text. These personas were manually reviewed by the authors to ensure they do not contain or infer any personally identifiable information or sensitive user data.

## REPRODUCIBILITY STATEMENT

We are committed to ensuring the reproducibility of our work. The source code for our ChronoPlay framework and the complete, instantiated benchmarks for all three games are included in the supplementary materials and will be publicly released upon publication. Detailed descriptions of our data processing pipeline, synthesis prompts, and experimental setup are provided in the main paper and appendices to facilitate replication. We acknowledge that due to the stochastic nature of the Large Language Models used for data synthesis and the random sampling of templates, executing our generation pipeline will not produce bit-for-bit identical benchmark instances on each run. However, we have taken measures to ensure the statistical and conceptual reproducibility of our results. We have conducted multiple generation runs and, as shown in our human evaluation of the synthesized data (Appendix D.5), the resulting benchmarks consistently meet a high-quality standard.

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

# A    DATA COLLECTION AND CONSTRUCTION DETAILS

This section provides additional details on the instantiation of our benchmarks, including the rationale for game selection and the specific hyperparameter settings used for partitioning the game timelines.

## A.1    GAME SELECTION AND DESCRIPTION

We chose three games with distinct characteristics to ensure our benchmark covers a diverse range of dynamic scenarios. For each game, we construct a authority knowledge base from official game wikis and update announcements. To model user interest, we collect a large corpus of player discussions from major communities [1].

- **Dying Light 2** was chosen as an example of a mature game with long-term support. Its knowledge base grows over a multi-year period through significant patches. Our data collection spans over three and a half years for this game (Jan '22 - Jul '25), during which we constructed a knowledge base from 310 wiki documents and 199 official update articles. Concurrently, we collected 10,478 community posts to model long-term user interest drift. For this game, we collected wiki data from its Fandom page[2] and update information from the official website.

- **Dune: Awakening** is our case study for a newly launched game. The game is built on the massive pre-existing lore of the Dune universe, creating an enormous knowledge base from day one. Its initial knowledge base is immense, built from 3,377 wiki documents and 43 update articles. To capture the rapid shifts in focus during its volatile launch window (Jun '25 - Aug '25), we analyzed 37,079 community posts. We sourced its wiki data from the Dune: Awakening Community Wiki[3] and gathered update notes from the game's official Steam platform page.

- **PUBG Mobile** was selected as a high-velocity live-service game. Its environment constantly changes due to frequent updates and regular seasonal events. Its high velocity is evidenced by the 60,632 community posts we collected over an 18-month period (Jan '24 - Jul '25). The knowledge base for this period is grounded in 142 wiki documents and 27 frequent official updates, providing a basis to evaluate RAG systems in a rapidly changing environment. We sourced wiki data from its Fandom page[4] and collected update information from the official game website.

## A.2    HYPERPARAMETER SETTINGS FOR PHASE PARTITIONING

The partitioning of each game's timeline into distinct phases is governed by our user interest drift detection mechanism, which relies on the following key hyperparameters:

- **Topic Importance Factor** ($\gamma$): In our topic-weighted JSD calculation, we set $\gamma = 1.5$. This value enhances the weight of more prominent topics in the distribution. This ensures that the drift detection is focused on significant, mainstream shifts in community discussion rather than statistical noise from niche, long-tail topics.

- **Drift Threshold** ($\lambda_{JSD}$): The threshold for flagging a significant drift is uniformly set to $\lambda_{JSD} = 0.001$. This sensitive value was chosen based on a conservative principle: for a benchmark designed to study performance volatility, it is more rigorous to risk over-segmenting the timeline with minor fluctuations than to risk overlooking a genuine shift in user interest. This ensures we capture the full dynamic nature of the environment.

- **Sliding Window Size** ($W$): The window size for monitoring the topic distribution of new community questions is tailored to the specific dynamics of each game. For *DL2*, with its longer update cycle and more gradual shifts in player focus, we use a wide window of $W = 6$ months. For *Dune*, to capture the rapid, day-by-day changes during its critical

---

[1]Mainly communities including Reddit:https://www.reddit.com, Discord: https://discord.com, and Twitch: https://www.twitch.tv

[2]https://dyinglight.fandom.com/wiki/Dying_Light_Wiki

[3]https://awakening.wiki/

[4]https://pubgmobile.fandom.com/wiki/PUBG_Mobile_Wiki

launch period, we use a very narrow window of $W = 5$ days. For *PUBG*, reflecting its regular seasonal updates and events, we use a medium window of $W = 2$ months.

Applying these tailored hyperparameters, our drift detection mechanism partitioned the timeline for each game into a series of distinct phases. We identified **5 phases** for *Dying Light 2*, **6 phases** for *Dune: Awakening*, and **7 phases** for *PUBG Mobile*. Figure 9 illustrates the evolution of the main community discussion topics across these detected phases for each game, visually confirming the significant interest drifts captured by our method.

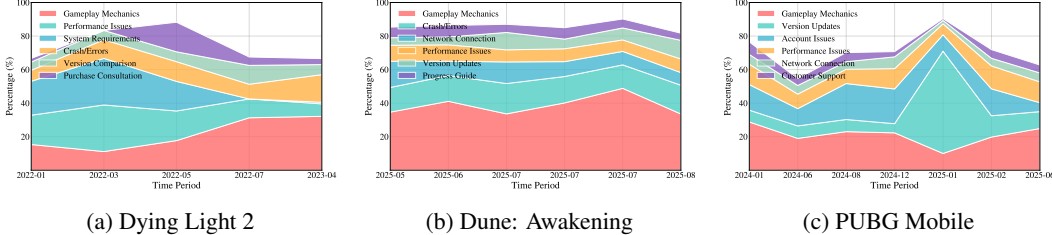

(a) Dying Light 2          (b) Dune: Awakening          (c) PUBG Mobile

Figure 9: Evolution of major user interest topics across the detected phases for each of the three games. Each colored area represents the proportion of a distinct discussion topic.

## B  HIERARCHICAL TOPIC TAXONOMY

To systematically categorize the vast number of user questions for analysis and benchmark generation, we developed a detailed hierarchical topic taxonomy. We first collected and sampled thousands of real user questions from the player communities of our three target games. Then, a team of domain experts manually grouped these questions into thematic clusters. Through several rounds of refinement and consolidation, these clusters were organized into the final hierarchical structure presented in Table 3. This taxonomy comprises 6 main categories and 21 sub-categories, covering the full spectrum of player inquiries from pre-purchase consultation to technical support and in-game strategies.

Table 3: The hierarchical topic taxonomy used for question classification.

| Main Category | Sub-category | Description |
|---|---|---|
| **Purchase Related** | Purchase Consultation | Inquiries about whether to buy, where to buy, and if the game is worth the price. |
| | Version Comparison | Questions about the differences between game versions. |
| | Platform Selection | Advice seeking on choosing between different platforms. |
| | Preorder Rewards | Inquiries about preorder bonuses, special rewards, or keys. |
| **Technical Support** | System Requirements | Questions regarding hardware specifications and compatibility. |
| | Performance Issues | Issues related to in-game performance, such as framerate, lag, and optimization. |
| | Crash & Errors | Problems like game crashes, freezes, black screens, or error messages. |
| | Network Connection | Issues with multiplayer connectivity, servers, or high latency. |
| **Game Content** | Gameplay Mechanics | Questions about basic controls, game systems, and how to play. |
| | Content & Features | Inquiries about the game's specific content, modes, world size, and story. |
| | Progress Guide | Questions seeking tips, guides, and advice on game progression. |
| | Version Updates | Questions about patches, updates, fixes, and new content. |
| **Social Interaction** | Team Cooperation | Inquiries related to finding teammates and playing with others. |
| | Friend System | Questions about the in-game social features, such as adding friends. |
| | Community Events | Questions about official or community-run events and competitions. |
| **After-sales Service** | Refund Policy | Questions regarding the process and conditions for getting a refund. |
| | Customer Support | Inquiries about how to contact customer service or report issues. |
| | Account Issues | Problems related to user accounts, login, or activation keys. |
| **Review & Discussion** | Review Questions | Questions about the game's ratings and reviews from others. |
| | Comparison Discussion | Discussions comparing the game to other similar titles. |
| | Expectation & Concern | Questions expressing hopes, worries, or concerns about the game's future. |

## C GENERATOR EVALUATION PROMPTS AND METRICS

This section provides the detailed methodology and LLM prompts used for evaluating the performance of the generator models. We assess the quality of generated answers based on two distinct criteria: correctness and faithfulness.

For both criteria, we employ a rigorous, 3-level scoring system, instructing the LLM judge to assign a score of 0, 1, or 2 based on a detailed rubric. **Score 2 (Excellent):** The answer is perfect or near-perfect according to the criterion. **Score 1 (Acceptable):** The answer is largely correct/faithful but has minor, non-critical flaws. **Score 0 (Defective):** The answer contains significant errors or fails to meet the core requirements of the criterion. To ensure consistency in our reporting, the final scores for each metric presented in the paper are normalized to a [0, 1] scale by dividing the raw score by the maximum possible score of 2.

### C.1 CORRECTNESS SCORE

Correctness measures the factual accuracy of the predicted answer when compared against a ground-truth answer. To ensure a strict and consistent evaluation, we provided the LLM judge with the following detailed prompt. **For the reader's clarity, we have organized the subsequent prompt's structure and bolded key headings and instructions, but this formatting was not present in the operational prompt used by our code.**

**Prompt for Correctness Evaluation**

---

You are an extremely strict game knowledge correctness evaluator. Your task is to evaluate the accuracy of a predicted answer against the ground truth answer for a game-related question with maximum rigor.

**ULTRA-STRICT EVALUATION CRITERIA:**

1. FACTUAL ACCURACY (ZERO TOLERANCE): Every single fact about game mechanics, rules, systems, and lore must be 100% correct. ANY factual error, no matter how minor, significantly impacts the score.

2. NUMERICAL INFORMATION (EXACT PRECISION): All numbers, statistics, values, quantities, percentages must be exactly correct. Even tiny numerical discrepancies are heavily penalized.

3. TERMINOLOGY AND NAMES (PERFECT ACCURACY): Character names, location names, item names, and ALL game-specific terms must be spelled exactly correctly.

4. COMPLETENESS AND COVERAGE (COMPREHENSIVE): The answer must address EVERY aspect of the question thoroughly. Missing ANY critical information from the ground truth is a major defect.

5. ADDITIONAL INFORMATION (STRICT VERIFICATION): Any extra information beyond the ground truth must be 100% accurate and verifiable. Speculative content or hallucinations result in immediate score reduction.

**ULTRA-STRICT 3-LEVEL SCORING SYSTEM:**

- 2 (Exceptionally Perfect): EVERY fact is 100% accurate. ALL numbers and terminology are precise. Comprehensively addresses the question. Truly exemplary answer.

- 1 (Acceptable with Minor Flaws): Core facts are accurate but contains 1-2 very minor issues (e.g., missing non-essential details).

- 0 (Defective/Inadequate): Contains ANY significant factual errors, multiple minor errors, notable numerical inaccuracies, or fails to address key aspects of the question.

**Question:** [Question]
**Retrieved Contexts:** [Documents]
**Predicted Answer:** [Answer]

Return your evaluation as a JSON object with the "accuracy" field (0, 1, or 2).

---

## C.2   FAITHFULNESS SCORE

Faithfulness measures whether the predicted answer is entirely grounded in and supported by the provided retrieved context documents. The prompt for Faithfulness is similarly strict, focusing on zero tolerance for hallucinations or any information not present in the provided context.

---

**Prompt for Faithfulness Evaluation**

---

You are an extremely strict faithfulness evaluator. Your task is to evaluate whether the predicted answer is entirely faithful to the provided retrieved context documents with MAXIMUM RIGOR.

**ULTRA-STRICT FAITHFULNESS EVALUATION CRITERIA:**

1. INFORMATION SOURCE VERIFICATION (ZERO TOLERANCE): EVERY piece of information in the predicted answer MUST be directly supported by the retrieved context. ANY claim not found in the context is a violation.
2. FACTUAL CONSISTENCY (EXACT ALIGNMENT): All facts, numbers, and names must match EXACTLY with the context. No paraphrasing that changes meaning.
3. CONTEXT GROUNDING (MANDATORY SUPPORT): Each statement must be traceable to specific parts of the retrieved contexts. No external knowledge is allowed.
4. HALLUCINATION DETECTION (ZERO TOLERANCE): Any information not present in the contexts is considered hallucination, even "common knowledge".
5. OMISSION vs ADDITION PRINCIPLE: Incomplete but faithful answers are preferred over complete but unfaithful ones.

**ULTRA-STRICT 3-LEVEL SCORING SYSTEM:**

- 2 (Perfectly Faithful): EVERY statement is directly supported by the retrieved contexts. No hallucinations, no external information.
- 1 (Mostly Faithful with Minor Issues): Core information is supported but contains 1-2 minor unsupported details or slight paraphrasing.
- 0 (Unfaithful/Hallucinated): Contains significant information not found in the contexts or any clear hallucination.

**Question:** [Question]
**Retrieved Contexts:** [Documents]
**Predicted Answer:** [Answer]

Return your evaluation as a JSON object with the "faithfulness" field (0, 1, or 2).

---

## C.3   VALIDATION OF LLM-AS-JUDGE AGAINST HUMAN EXPERTS

To validate the reliability of our LLM-as-Judge approach, we conducted an experiment to measure its agreement with human expert evaluations. For this validation, we used GPT-4o as our LLM judge. We randomly sampled 150 question-answer pairs (50 from each of the three games) generated by a representative RAG system using text-embedding-3-small for retrieval and GPT-4o for generation. Three domain experts were recruited to perform a binary evaluation (0: Fail, 1: Pass) for both correctness and faithfulness, and their majority vote was treated as the ground truth. To streamline the process, a web-based annotation interface was provided to the experts, as shown in Figure 10.

To compare the 3-level scores from the LLM with the binary human scores, we applied a lenient mapping: LLM scores of 2 ("Excellent") and 1 ("Acceptable") were mapped to 1 (Pass), while a score of 0 ("Defective") was mapped to 0 (Fail). We then assessed the alignment using standard classification metrics. The results are presented in Table 4.

Table 4: Performance of the LLM-as-Judge against the human expert ground truth. The high precision indicates the model is a reliable, albeit strict, evaluator.

| Criterion | Accuracy | Precision | Recall | F1-Score |
|---|---|---|---|---|
| Correctness | 70.67% | 98.77% | 65.04% | 78.43% |
| Faithfulness | 78.00% | 96.30% | 78.20% | 86.31% |

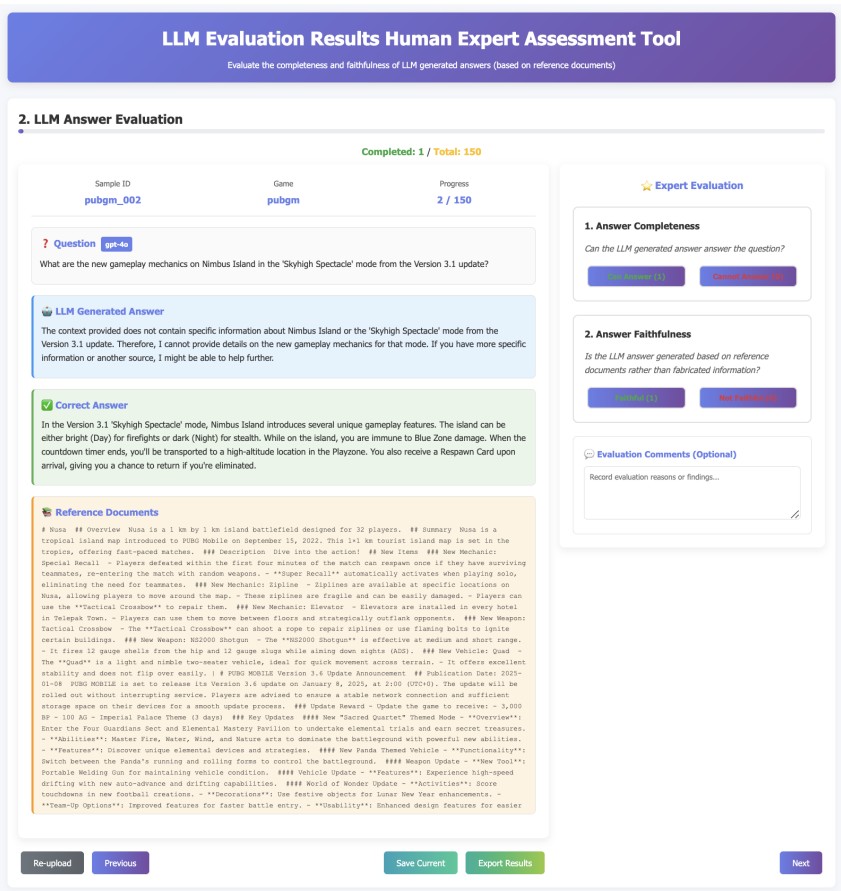

Figure 10: The web interface provided to human experts for the binary evaluation of correctness and faithfulness.

The results reveal key insights into the behavior of our automated judge. The most notable finding is the extremely high precision for both correctness (98.77%) and faithfulness (96.30%). This indicates that the LLM judge rarely commits a "false positive" error. When LLM classifies an answer as high-quality (Pass), we can be very confident in that judgment. Conversely, the model shows more conservative recall (65.04% for correctness), meaning it is stricter than human experts and sometimes flags acceptable answers as "Fail". This is a deliberate and desirable characteristic for our benchmark's automated judge. In a high-stakes domain like gaming, where an incorrect guide or faulty information can directly waste a player's time and ruin their experience, prioritizing precision over recall is crucial. A strict, conservative judge ensures that only truly high-quality and reliable answers receive a passing score. This guarantees the integrity and high standard of our benchmark, ensuring that models that perform well are genuinely robust. Overall, with acceptable accuracy and high F1-scores, these results validate our use of an LLM, guided by our strict prompts, as a scalable and reliable method for evaluating generator performance.

## D TECHNICAL DETAILS OF THE CHRONOPLAY FRAMEWORK

This section provides a description of the key components of the ChronoPlay, covering the methods used for NER, the extraction of community-driven data assets, and the agent-based data synthesis pipeline. The entire synthesis pipeline employs the GPT-4o model. Furthermore, we conducted a human expert evaluation on the synthesized data to validate the quality of the synthesis.

## D.1 NAMED ENTITY RECOGNITION VIA SELF-ICL

A core requirement for our dynamic update mechanism is to accurately identify in-game entities $\sigma$ within the knowledge base. To avoid the need for extensive manual annotation for each new game, we employ a NER strategy based on Self-ICL (Chen et al., 2023). This process works in three main steps:

- **Pseudo-Input Generation:** Given a target text for annotation, we first use an LLM to generate several text samples that are stylistically similar but feature diverse content. These pseudo-inputs are designed to maintain the same domain characteristics and linguistic patterns as the original text, effectively creating a set of rich, in-domain examples.

- **Pseudo-Label Prediction:** Next, we use a zero-shot prompt to perform an initial entity annotation on the pseudo-inputs generated in the previous step. This predicts the likely in-game entities and their types for each pseudo-input, resulting in a collection of (pseudo-input, pseudo-label) pairs that serve as noisy but relevant training demonstrations.

- **In-Context Learning Stage:** Finally, these (pseudo-input, pseudo-label) pairs are used as demonstrations for in-context learning. We construct a new prompt that includes these concrete examples, followed by the original, real target text. Guided by these demonstrations, the LLM performs a more accurate and context-aware entity recognition on the real text, having learned from the provided examples.

This Self-ICL method offers significant advantages over traditional zero-shot NER, primarily through superior domain adaptation. By generating pseudo-examples specific to the gaming context, the model effectively learns the unique linguistic patterns and expressions of the domain. This allows for a more powerful form of in-context learning, where the model is guided by concrete example-label pairs rather than abstract definitions. The prompts for the three stages described above are detailed below.

**Prompt 1: Pseudo-Input Generation**

---

You are tasked with generating pseudo-inputs for game-related Named Entity Recognition.

Given the following game-related text, generate [Num_Pseudo_Examples] similar but different game-related text examples that would be suitable for entity recognition. The generated texts should: 1. Be similar in style and domain to the input text 2. Contain various types of game entities 3. Be realistic and coherent 4. Have different specific entities but similar context patterns

Original text: [Question]

Return the result in the following JSON format only, no other text:

```
{{
    "pseudo_inputs": [
        "example text 1",
        "example text 2",
        "example text 3"
    ]
}}
```

---

**Prompt 2: Pseudo-Label Prediction**

---

Extract game-related entities from the following text.

Entity Types: [Entity_Desc]

Text: [Pseudo_Text]

Return the result in the following JSON format only, no other text:

```
{{
    "entities": [
        {{
            "text": "entity text",
            "type": "ENTITY_TYPE",
```

```
            "context": "brief context"
        }}
    ]
}}
```

---

**Prompt 3: In-Context Learning**

---

You are a specialized assistant for identifying game-related entities. Learn from the following examples and then extract entities from the test input.

Entity Type Definitions: [Entity_Desc]

Here are some examples: [Demonstrations_Text]

Now, extract entities from the following test input.

Test Input: [Question]

Return the result in the following JSON format only, no other text:

```
{{
    "entities": [
        {{
            "text": "entity text",
            "type": "ENTITY_TYPE",
            "context": "brief context"
        }}
    ]
}}
```

---

## D.2 EXTRACTION OF QUESTION TEMPLATES ($\mathcal{T}_{comm}$) AND USER PERSONAS ($\mathcal{U}_{comm}$)

To capture the authenticity of user questions as described in Section 3.1.2, we developed a pipeline to process data collected from player communities. The pipeline first filters posts to acquire genuine user questions. Each filtered question is then passed through an LLM classifier, which assigns it a topic $\theta$ from our hierarchical taxonomy (see Appendix B). Finally, a subsequent LLM prompt is used to decouple and extract two core, reusable assets from each classified question: question templates $p$ and their associated user personas $u$. The deduplicated collections of these assets form the Question Template Base $\mathcal{T}_{comm}$ and the User Persona Base $\mathcal{U}_{comm}$.

**Question Template Extraction**: The extraction of the Question Template Base $\mathcal{T}_{comm}$ aims to capture authentic, game-agnostic query patterns. The process is orchestrated by an LLM instructed to perform a generalization task. For each user question, the model identifies and anonymizes specific in-game entities (e.g., character names, item names) by replacing them with standardized placeholders (e.g., [CHARACTER_NAME]). This transformation results in a set of abstract question templates $p$. Each template is then associated with its corresponding topic $\theta$ from our taxonomy (see Appendix B). The resulting template-topic pairs $(p, \theta)$ are stored in the Question Template Base. This decoupling of query patterns from specific game instances is crucial for the framework's scalability.

---

**Prompt for Question Template Extraction**

---

Please analyze the following question and generate 2-3 abstract question templates.

**Question Information:**

- **Question:** [Question_Content]
- **Question Topic:** [Question_Topic]

**Generation Requirements:**

1. Replace specific game names, platform names, character names, etc., with placeholders.
2. Maintain the core structure and intent of the question.
3. Each template should be usable for generating similar questions.
4. Use placeholder format: `[PLACEHOLDER_NAME]`.

**Common Placeholders:**

- `[GAME_NAME]`: Game name
- `[PLATFORM]`: Platform name (Steam, Xbox, Epic, etc.)
- `[CHARACTER_NAME]`: Character name
- And other relevant placeholders like `[ITEM]`, `[LOCATION]`, `[VERSION]`, etc.

**Output Format:** Please return in JSON format, containing a templates array, with each template including: a template text, a list of placeholders, and a description. Example format:

```
{
  "templates": [
    {
      "template": "Does pre-ordering on [PLATFORM1] also give
      [REWARD_TYPE], and is there any difference from [PLATFORM2]?",
      "placeholders": ["PLATFORM1", "PLATFORM2", "REWARD_TYPE"],
      "description": "Template for asking about pre-order reward
      differences between platforms"
    },
    {
      "template": "What's the difference between pre-order rewards
      on [PLATFORM] and [ANOTHER_PLATFORM]?",
      "placeholders": ["PLATFORM", "ANOTHER_PLATFORM"],
      "description": "Template for comparing pre-order rewards
      across platforms"
    }
  ]
}
```

**User Persona Extraction**: Concurrently with template extraction, the same LLM prompt instructs the model to analyze the user's language, tone, and the context of their query to infer a plausible user persona $u$. The output is a concise narrative description (e.g., "A new player struggling with the crafting system..."). To ensure the quality of the User Persona Base $\mathcal{U}_{comm}$, the generation process is guided by several constraints. The persona must be inferred solely from the provided text to prevent hallucination. Each generated persona is also accompanied by a model-generated confidence score, and questions lacking sufficient context for a high-confidence inference are discarded.

**Prompt for User Persona Extraction**

Please analyze the following gaming player's question and generate a concise player background description.

**Question:** [Question_Content]

Based on the question content, write a 50-100 word player background description that describes the player's gaming experience, skill level, interests, and preferences.

**Requirements:**

1. The description should be natural and fluent, like a brief introduction of a person.
2. Only make reasonable inferences based on the question content; do not over-interpret.
3. If the question is too simple or contains no personal information, return null.
4. Use a second-person ("You are a player who...") description format.

**Output Format:** Please return in JSON format. Example of a successful extraction:

```
{
  "player_description": "You are a player who...",
  "confidence_score": 0.8
}
```

If no meaningful player information can be inferred, please return:

```
{
  "player_description": null,
  "confidence_score": 0.0
}
```

**Semantic Deduplication**: As a final quality control step, we employ a vector-based filtering mechanism to deduplicate semantically similar items in both the $\mathcal{T}_{comm}$ and $\mathcal{U}_{comm}$ bases. We use a sentence-embedding model (text-embedding-3-small) to convert assets into vector representations and compute their cosine similarity. Pairs with a similarity score exceeding a predefined threshold $0.7$ are flagged as duplicates and filtered out. The deduplicated sets then undergo a final manual review by domain experts to ensure the high quality and diversity of the final asset bases.

### D.3 HYPOTHETICAL Q&A GENERATION FOR IMPROVED RETRIEVAL

To bridge the semantic gap between an abstract question template and a detailed knowledge document, we employ a hypothetical question-answer (Q&A) generation step inspired by techniques like HyDE (Gao et al., 2023). This process is designed to create a semantically rich query vector that significantly enhances the relevance of documents retrieved for synthesis.

The core of this process is a single LLM call that generates a complete, hypothetical Q&A pair directly from a question template. Given a template from our $\mathcal{T}_{comm}$ base, its associated topic, and the target game name, we use a specialized prompt to instruct an LLM. The prompt guides the model to perform two actions in one step. First, it instantiates the abstract template into a specific, plausible question by filling in its placeholders. Second, it generates a detailed, hypothetical answer to this newly created question. The model fabricates this answer based on its general world knowledge of game-like structures, without access to our specific knowledge base. The embedding of this semantically rich, hypothetical question and answer are then used as the query vector to perform a search against the knowledge base $\mathcal{K}_{auth}$.

---

**Prompt for Hypothetical Q&A Generation**

---

Based on the following question template, generate a specific question and corresponding hypothetical answer. Please ensure that placeholders in the template are replaced with appropriate content. Pay special attention to the game placeholder `[GAME_NAME]`, please use the correct game name *game_name*.

**Question Template**: [Question_Template]

**Question Topic**: [Question_Topic]

Please provide your response strictly in the following JSON format. Do not include any other text.

```
{
  "question": "The specific question you generated based on
  the template.",
  "answer": "The helpful and reasonable hypothetical answer
  to the question."
}
```

**Requirements**:

1. Questions should be specific and clear, conforming to gaming players' expression habits.
2. Answers should be reasonable and helpful.
3. If there are placeholders in the template (such as `[GAME_NAME]`), please replace them with appropriate content.
4. The answer's length should be moderate, both useful and not too lengthy.

---

### D.4 DATA SYNTHESIS AGENT

The entire data synthesis process is orchestrated by a Data Synthesis Agent. This agent executes a multi-stage workflow that fuses multiple data assets to generate candidate question-answer pairs, which are then immediately evaluated by a real-time quality control mechanism to ensure only high-fidelity data is included in the final benchmark.

#### D.4.1 MULTI-STAGE SYNTHESIS PROCESS

The agent follows a multi-stage synthesis process that fuses multiple data assets. It combines a sampled user persona and question template with pre-processed entities and specific task requirements

(e.g., single-hop vs. multi-hop reasoning) to create a comprehensive prompt. This prompt, detailed below, guides a powerful LLM to generate a candidate Q&A pair.

**Prompt for Data Synthesis Agent (Data Generation)**

---

**SYSTEM PROMPT**:

**Background**: You are an intelligent evaluation data generation assistant with deep role-playing capabilities. I am building a multi-task evaluation dataset for retrieval-augmented gaming large language models. I require you to automatically generate gaming domain evaluation data that is strongly correlated with evaluation tasks. I will provide the following content: [gaming subtopics of focus for evaluation data, task descriptions and requirements for evaluation, gaming documents from the knowledge base, and possible player role backgrounds]. You need to generate high-quality evaluation data based on the following principles:

1. **Role Consistency**: If a player role background is provided, you must fully immerse yourself in that role's identity and language style.
2. **Authenticity Simulation**: Generated questions must reflect real players' questioning habits and expression patterns.
3. **Personalized Expression**: Questions from different roles should reflect different gaming experience levels and focus points.
4. **Template Guidance**: If question templates are provided, generated questions should reference the template's structure and style, but with reasonable variations.

**Quality Requirements for Generated Data**

- **For Documents:** Must be high-quality gaming materials (official guides, update logs, etc.). Do not generate from low-quality or private content. If irrelevant, return an empty list.
- **For Questions:** Must be role-driven, specific, semantically complete, and strongly related to the topic. Must not contain phrases like "according to the document." Must strictly comply with the task definition (e.g., multi-hop reasoning).
- **For Answers:** Must have high knowledge density, be factually consistent with the document, and contain no hallucinations.
- **For References:** Must accurately extract segments from the original document that support the answer. Extracted content must be informationally complete and not taken out of context.

**Data Generation Process:** 1. First, determine if the provided document is high-quality and relevant to the task. If not, return an empty list. 2. If suitable, generate high-quality evaluation samples based on the document content, task requirements, and gaming topics.

**Generated Data Format Requirements**

First start with ###THOUGHT_PROCESS###, output your thinking process, then output results in a JSON data list format, surrounded by `<json></json>`. The format requirements are as follows:

```
<json>
[
    {
        "question": "Question raised from player perspective...",
        "answer": "Direct answer to the question,
        concise and clear...",
        "references": ["Specific content segment 1...",
        "Quote segment 2"]
    }
]
</json>
```

**USER PROMPT**:

**Gaming Subtopics of Focus for Evaluation Data** [Topic_Description]

**Evaluation Task Description and Requirements**

Query Type: [Query_Type]: [Query_Type_Description]

Role Context: [Role_Context]

Question Template: [Question_Template]

**Question Generation Specificity Guidelines**

**Important Reminder**: Generated questions must be sufficiently specific.

- **Hardware-related**: Must specify exact models (e.g., "RTX 4070").
- **Location-related**: Must specify exact area names (e.g., "Harran City Center").
- **Numerical-related**: Must provide specific values or ranges (e.g., "level 30 and above").
- **Game Content**: Must use accurate game terminology.
- **Version Handling**: Do not directly mention version numbers in questions.

**Provided Document**
[Documents]

---

### D.4.2 REAL-TIME QUALITY CONTROL MECHANISM

Immediately after synthesis, each candidate Q&A pair is passed to a real-time quality control module. This module uses an LLM, guided by the prompt below, to score the data's quality on a three-point scale (0: poor, 1: average, 2: excellent) based on criteria like task compliance and answer accuracy. Only data scoring 2 is retained.

**Prompt for Quality Control Agent**

---

**SYSTEM PROMPT**:

**Background**: You are a professional generated data quality assessor. I will provide you with evaluation data generated by a large language model. Your task is to assess the quality of this data. The quality is divided into three levels: 0 (poor), 1 (average), 2 (high quality).

**Assessment Requirements**

1. Determine if generated questions are related to the provided gaming subtopics.
2. Determine if questions meet the requirements of the evaluation subtask.
3. Determine if answers are correct and can be fully answered by the long document.
4. Determine if the relevant segments are complete and sufficient to support the answer.

**Special Attention Points:**

- For "yes/no" questions, please mark their quality as 0.
- For multi-hop reasoning Q&A, please ensure the question requires at least two steps of reasoning. If it is a pseudo multi-hop question, its quality should be 0 or 1.

**Output Requirements**
Assessment results should be returned in JSON format: `{"evaluation": [0,1,2]}`
**USER PROMPT**:
**Long Document in Gaming Domain Used to Generate Data**: [Documents]
**Gaming Subtopics that Generated Data Should Conform To**: [Topic_Description]
**Description and Requirements of Evaluation Subtasks**: [Query_Type]
**Evaluation Data Generated by Large Language Model to be Assessed**: [Generated_Data]

---

### D.5 HUMAN EXPERT EVALUATION OF SYNTHESIZED DATA

To validate the quality of the data generated by our synthesis pipeline, we conducted a comprehensive evaluation with human domain experts. We recruited three experts, all of whom are veteran players of the target games. A random sample of 210 instances (70 from each game) was selected for this evaluation.

The experts performed the annotation using a web interface, designed to facilitate a clear and efficient evaluation process, as shown in Figure 11. To ensure consistent and high-quality annotations, we provided all experts with a detailed annotation guideline. The guideline instructed annotators to perform a binary (Yes/No) evaluation on three dimensions for each data sample: **(1) Correctness**, assessing if the answer correctly addresses the question; **(2) Reference Quality**, assessing if the cited documents are relevant; and **(3) Entity Accuracy**, assessing if the extracted entities are correct. The guideline emphasized a strict evaluation for Correctness and Reference Quality. In contrast, it instructed a more lenient approach for Entity Accuracy, where only clear and significant errors would result in a "No." This lenient standard was chosen because a sample can contain many potential entities, and what constitutes a relevant entity can be subjective. A stricter requirement could lead to large annotation variances due to differing expert interpretations and preferences.

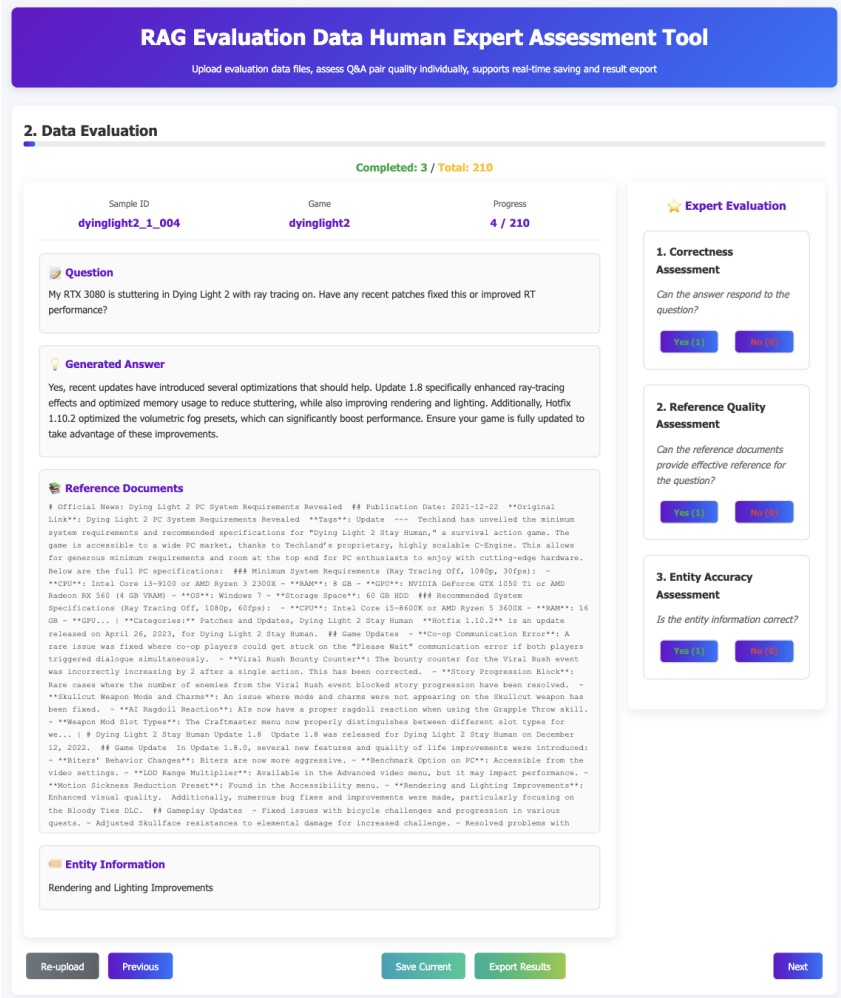

Figure 11: The web interface for human expert evaluation. Annotators can view the question, answer, reference documents, and entities in one place to provide scores for each dimension.

The average scores from the three experts across the three dimensions are presented in Table 5. We also calculated the inter-rater reliability among the three annotators using Krippendorff's Alpha, a standard measure of agreement.

Table 5: Human expert evaluation results. Scores represent the average of three experts on a binary (0/1) scale. Agreement is measured using Krippendorff's Alpha Ford (2004).

| Game | Correctness | Reference Quality | Entity Accuracy | Agreement |
|------|-------------|-------------------|-----------------|-----------|
| Dune: Awakening | 0.973 | 0.956 | 0.929 | 0.952 |
| PUBG Mobile | 0.932 | 0.971 | 0.914 | 0.934 |
| Dying Light 2 | 0.923 | 0.963 | 0.914 | 0.923 |

The results show a very high level of quality for our synthesized data across all evaluated dimensions, with average scores consistently above 0.91 for all games. The high inter-rater agreement further validates the clarity of our annotation guideline and the reliability of the evaluation results, confirming that our data generation pipeline produces high-quality and trustworthy benchmark data.

# E SUPPLEMENTAL RETRIEVAL RESULTS FOR K=1 AND K=5

This section provides supplemental results for the retriever evaluation to offer a more comprehensive view of model performance. Table 6 and Table 7 present the detailed, phase-by-phase results when the number of retrieved documents is set to 1 and 5, respectively. These tables complement the K=3 results shown in the main paper.

Table 6: Retrieval performance on all three games for **K=1**. We report Recall@1 (R@1), F1@1, and NDCG@1 (N@1). The best performing result in each row is in **bold**.

| Phase | BM25 | | | Qwen3-Embedding | | | BGE-M3 | | | text-embedding-3 | | |
|---|---|---|---|---|---|---|---|---|---|---|---|---|
| | R@1 | F1@1 | N@1 | R@1 | F1@1 | N@1 | R@1 | F1@1 | N@1 | R@1 | F1@1 | N@1 |
| *DL2* | | | | | | | | | | | | |
| 1 | 0.270 | 0.383 | 0.718 | 0.219 | 0.309 | 0.578 | 0.239 | 0.339 | 0.637 | **0.291** | **0.416** | **0.787** |
| 2 | 0.299 | 0.419 | 0.777 | 0.213 | 0.291 | 0.522 | 0.238 | 0.331 | 0.608 | **0.305** | **0.430** | **0.802** |
| 3 | 0.303 | 0.426 | 0.792 | 0.250 | 0.346 | 0.630 | 0.263 | 0.367 | 0.677 | **0.315** | **0.443** | **0.828** |
| 4 | 0.226 | 0.322 | 0.608 | 0.188 | 0.264 | 0.492 | 0.188 | 0.265 | 0.495 | **0.247** | **0.352** | **0.665** |
| 5 | 0.204 | 0.290 | 0.547 | 0.164 | 0.228 | 0.420 | 0.177 | 0.250 | 0.465 | **0.262** | **0.375** | **0.713** |
| *Dune* | | | | | | | | | | | | |
| 1 | 0.194 | 0.282 | 0.540 | 0.237 | 0.344 | 0.656 | 0.000 | 0.000 | 0.000 | **0.240** | **0.348** | **0.668** |
| 2 | 0.188 | 0.270 | 0.506 | 0.211 | 0.303 | 0.568 | 0.000 | 0.000 | 0.000 | **0.236** | **0.341** | **0.646** |
| 3 | 0.238 | 0.341 | 0.640 | 0.257 | 0.367 | 0.688 | 0.000 | 0.000 | 0.000 | **0.272** | **0.391** | **0.738** |
| 4 | 0.188 | 0.269 | 0.504 | 0.215 | 0.308 | 0.580 | 0.000 | 0.000 | 0.000 | **0.230** | **0.332** | **0.630** |
| 5 | 0.179 | 0.254 | 0.470 | 0.219 | 0.314 | 0.588 | 0.000 | 0.000 | 0.000 | **0.232** | **0.333** | **0.624** |
| 6 | 0.181 | 0.262 | 0.502 | 0.201 | 0.293 | 0.562 | 0.000 | 0.000 | 0.000 | **0.225** | **0.327** | **0.630** |
| *PUBG* | | | | | | | | | | | | |
| 1 | 0.273 | 0.410 | 0.820 | 0.258 | 0.388 | 0.775 | **0.278** | **0.417** | **0.835** | 0.259 | 0.393 | 0.782 |
| 2 | 0.158 | 0.237 | 0.475 | 0.158 | 0.237 | 0.475 | **0.160** | **0.240** | **0.480** | 0.157 | 0.235 | 0.470 |
| 3 | 0.245 | 0.367 | 0.735 | 0.255 | 0.383 | 0.765 | **0.257** | **0.385** | **0.770** | 0.245 | 0.367 | 0.735 |
| 4 | 0.145 | 0.217 | 0.435 | 0.153 | 0.230 | 0.460 | 0.158 | 0.237 | 0.475 | **0.163** | **0.245** | **0.490** |
| 5 | **0.240** | **0.360** | **0.720** | 0.218 | 0.328 | 0.655 | 0.222 | 0.333 | 0.665 | 0.200 | 0.300 | 0.600 |
| 6 | 0.233 | 0.350 | 0.700 | 0.232 | 0.347 | 0.695 | **0.235** | **0.352** | **0.705** | 0.233 | 0.350 | 0.700 |
| 7 | 0.202 | 0.302 | 0.605 | 0.212 | 0.318 | 0.635 | **0.220** | **0.330** | **0.660** | 0.208 | 0.312 | 0.625 |

Table 7: Retrieval performance on all three games for **K=5**. We report Recall@5 (R@5), F1@5, and NDCG@5 (N@5). The best performing result in each row is in **bold**.

| Phase | BM25 | | | Qwen3-Embedding | | | BGE-M3 | | | text-embedding-3 | | |
|---|---|---|---|---|---|---|---|---|---|---|---|---|
| | R@5 | F1@5 | N@5 | R@5 | F1@5 | N@5 | R@5 | F1@5 | N@5 | R@5 | F1@5 | N@5 |
| *DL2* | | | | | | | | | | | | |
| 1 | 0.439 | 0.307 | 0.528 | 0.426 | 0.299 | 0.524 | 0.454 | 0.320 | 0.571 | **0.607** | **0.433** | **0.807** |
| 2 | 0.422 | 0.291 | 0.529 | 0.385 | 0.263 | 0.435 | 0.417 | 0.290 | 0.492 | **0.615** | **0.436** | **0.790** |
| 3 | 0.447 | 0.309 | 0.553 | 0.406 | 0.278 | 0.508 | 0.430 | 0.299 | 0.551 | **0.636** | **0.450** | **0.844** |
| 4 | 0.366 | 0.259 | 0.456 | 0.323 | 0.228 | 0.457 | 0.362 | 0.255 | 0.482 | **0.502** | **0.357** | **0.711** |
| 5 | 0.337 | 0.237 | 0.411 | 0.312 | 0.219 | 0.423 | 0.335 | 0.236 | 0.437 | **0.505** | **0.359** | **0.678** |
| *Dune* | | | | | | | | | | | | |
| 1 | 0.334 | 0.237 | 0.417 | 0.399 | 0.286 | 0.666 | 0.167 | 0.122 | 0.128 | **0.434** | **0.313** | **0.709** |
| 2 | 0.329 | 0.232 | 0.404 | 0.381 | 0.271 | 0.628 | 0.066 | 0.049 | 0.058 | **0.394** | **0.280** | **0.693** |
| 3 | 0.347 | 0.244 | 0.465 | 0.399 | 0.280 | 0.729 | 0.084 | 0.062 | 0.073 | **0.427** | **0.303** | **0.785** |
| 4 | 0.313 | 0.220 | 0.387 | 0.367 | 0.261 | 0.615 | 0.083 | 0.061 | 0.078 | **0.394** | **0.281** | **0.664** |
| 5 | 0.307 | 0.214 | 0.378 | 0.371 | 0.261 | 0.634 | 0.077 | 0.057 | 0.074 | **0.384** | **0.272** | **0.675** |
| 6 | 0.302 | 0.215 | 0.377 | 0.369 | 0.265 | 0.605 | 0.107 | 0.080 | 0.094 | **0.387** | **0.280** | **0.652** |
| *PUBG* | | | | | | | | | | | | |
| 1 | 0.478 | 0.359 | 0.540 | **0.595** | **0.446** | 0.630 | 0.555 | 0.416 | 0.615 | **0.595** | **0.446** | **0.634** |
| 2 | 0.308 | 0.231 | 0.334 | **0.445** | **0.334** | **0.446** | 0.420 | 0.315 | 0.425 | 0.440 | 0.330 | 0.439 |
| 3 | 0.582 | 0.436 | 0.605 | **0.680** | **0.510** | **0.684** | 0.642 | 0.481 | 0.656 | 0.678 | 0.509 | 0.674 |
| 4 | 0.287 | 0.215 | 0.309 | **0.390** | **0.292** | 0.402 | 0.365 | 0.274 | 0.385 | **0.390** | **0.292** | **0.410** |
| 5 | 0.380 | 0.285 | 0.445 | **0.453** | **0.340** | **0.498** | 0.437 | 0.328 | 0.483 | 0.423 | 0.318 | 0.461 |
| 6 | 0.408 | 0.306 | 0.459 | 0.567 | 0.425 | 0.587 | 0.510 | 0.383 | 0.545 | **0.580** | **0.435** | **0.597** |
| 7 | 0.393 | 0.295 | 0.427 | **0.537** | **0.403** | **0.564** | 0.483 | 0.362 | 0.527 | 0.512 | 0.384 | 0.543 |

As shown in Table 6 and Table 7, the key trends observed for K=3 in the main analysis are consistent across K=1 and K=5 settings. The summarized conclusions are as follows:

- **Performance Volatility:** The performance of all models continues to exhibit significant volatility across the lifecycle phases. The performance dip in Phase 4 of *PUBG* is clearly

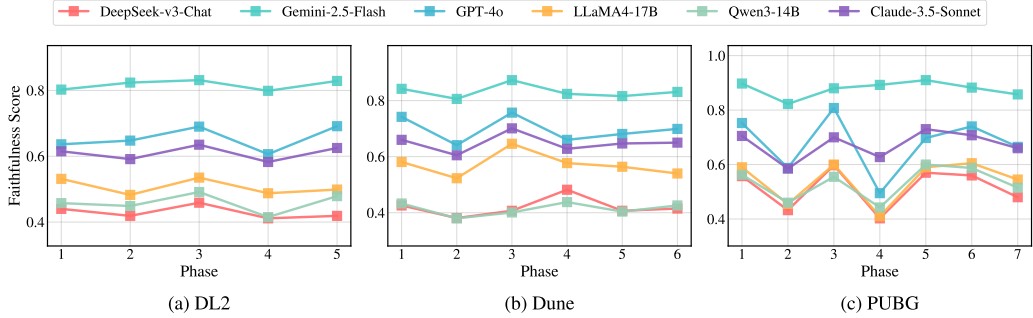

Figure 12: Generator faithfulness scores across the lifecycle phases of each game.

visible in both the K=1 and K=5 results, confirming that major game events impact retrieval regardless of the number of documents returned.

- **Model Rankings:** The relative ranking of the models also remains largely consistent. text-embedding-3 generally maintains its lead, but other models like Qwen3-Embedding and BGE-M3 remain competitive in specific phases.

- **Domain-Specific Challenges:** Notably, the extremely low performance of BGE-M3 on the *Dune: Awakening* benchmark persists across all K values, reinforcing our hypothesis that this model struggles with the unique, noun-heavy vocabulary of the Dune universe.

As expected, increasing K from 1 to 5 generally improves Recall and F1 scores for all models, as there is a higher chance of retrieving a relevant document. However, the overall performance patterns and the challenges posed by dynamic shifts remain the same. This demonstrates the robustness of our findings and validates that the challenges identified by our benchmark are fundamental and not an artifact of the specific value of K.

## F    GENERATION PERFORMANCE ON FAITHFULNESS SCORES

This section provides the detailed faithfulness scores for the generator evaluation, complementing the correctness scores presented in the main paper. Faithfulness measures whether the generated answer is grounded in and strictly supported by the provided context documents. The analysis of Faithfulness scores in Figure 12 reveals several unique insights into model behavior that complement the Correctness results from the main paper.

First, we observe a shift in the top-performing model. While GPT-4o excelled in correctness, Gemini-2.5-flash consistently achieves the highest faithfulness scores across all games and phases. This suggests that different models exhibit different strengths. Some are better at producing factually correct answers, while others are better at strictly adhering to the provided documents.

Second, the results show that faithfulness is not always positively correlated with correctness. This divergence highlights a fundamental trade-off in RAG systems. A model can achieve high faithfulness by accurately repeating information from a retrieved document, but if that document is incorrect or outdated, the final answer will have low correctness. Conversely, a model might use its internal parametric knowledge to produce a correct answer even when the retrieved context is insufficient, leading to high correctness but low faithfulness. Our benchmark's ability to measure both dimensions is crucial for identifying these different model behaviors.

Finally, just like correctness, faithfulness is strongly influenced by the benchmark's dynamic nature. The significant fluctuations, particularly for *PUBG*, demonstrate that a model's tendency to hallucinate or generate ungrounded responses is not a fixed trait. Instead, it is highly dependent on the challenges of each lifecycle phase, such as the quality of retrieved documents. This analysis confirms that a comprehensive evaluation requires a dynamic benchmark that can surface these complex performance trade-offs over time.

# G    Topic Performance Analysis

To provide a more fine-grained understanding of the RAG system's performance in the game domain, we analyzed it on a topic basis. For this evaluation, we used a RAG system with text-embedding-3 as the retriever and GPT-4o as the generator. Figure 13 presents a heatmap that visualizes the performance across the top six most frequent topics for each of the three games. It is important to note that the dominant topics vary significantly between games, reflecting the unique focus and lifecycle stage of each player community. This analysis allows us to identify which types of user interests pose the greatest challenges to modern RAG systems.

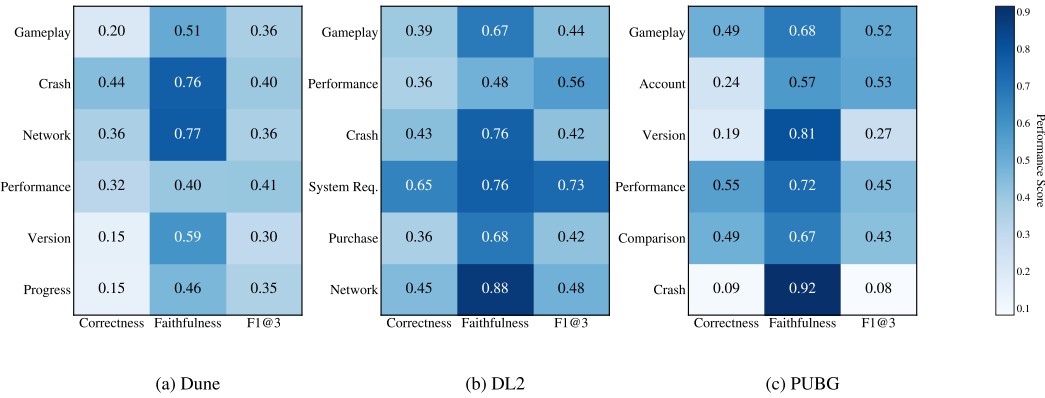

Figure 13: Heatmap of RAG performance across the top six topics for each game. Lighter colors indicate lower performance, highlighting challenging topics for the RAG system.

The topic performance breakdown reveals several key insights. First, performance is not uniform across topics. Technical, fact-based topics with clear answers generally perform well. For example, the *SYSTEM_REQUIREMENTS* topic in *DL2* achieves high scores across retrieval, correctness, and faithfulness. In contrast, more nuanced and complex topics consistently challenge the RAG system. This analysis provides a deeper, data-driven explanation for the performance drop observed in our main analysis (Section 4.2.1). For *DL2*, the *GAMEPLAY_MECHANICS* topic, which became dominant during the challenging Phase 4, shows a significantly lower Retrieval F1 score (0.44) compared to more straightforward topics like *SYSTEM_REQUIREMENTS* (0.73). This confirms that the performance degradation was caused by a shift in user interest towards an inherently more difficult topic.

Furthermore, the topic performance breakdown highlights the critical divergence between faithfulness and correctness, revealing a key failure mode for RAG systems. For instance, on the *CRASH_ERRORS* topic in *PUBG*, the system achieves a very high faithfulness score (0.92) but an extremely low correctness score (0.09). Crucially, this occurs in a scenario where the retrieval performance is also exceptionally poor (F1@3 of 0.08). This paradoxical outcome indicates a specific failure cascade: the retriever is failing to find the correct documents (e.g., official solutions or patch notes). Instead, it is likely retrieving topically similar but factually incorrect documents. The generator then performs its task perfectly, faithfully summarizing this incorrect information. The result is an answer that is well-grounded in the retrieved context but completely wrong. This underscores the importance of evaluating all components of the RAG pipeline, as a failure in retrieval can directly lead the generator to produce confident but misleading answers.

# H    Ablation Study Details and Clarity Analysis

This section provides a detailed description of the experimental setup for the ablation study presented in RQ3, along with an in-depth analysis of the evaluation results for the clarity criterion.

## H.1 EVALUATION CRITERIA AND METHODOLOGY

To assess the quality of the questions generated by our different synthesis pipelines, we defined two key criteria:

- **Authenticity:** This metric assesses how closely a generated question resembles a question a real human player would post in a community forum. It considers the tone, phrasing, use of domain-specific jargon, and the underlying user intent.
- **Clarity:** This metric evaluates how well-formed, unambiguous, and easy to understand the question is. It assesses grammatical correctness and the straightforwardness of the query.

Our evaluation employs a 4-way forced-choice comparison. To conduct this study, we randomly sampled 200 source documents for each of the three games, resulting in a total of 600 evaluation instances and 2,400 questions. For each instance, we generated four questions from each pipeline configuration. Then we presented them in a randomized order to the LLM. The LLM was tasked with selecting the single best question for each of the two criteria separately. The final score for each setting is its win rate: the percentage of times it was chosen as the best.

## H.2 LLM-AS-JUDGE PROTOCOL

We used a panel of three models as LLM judges: GPT-4o, Gemini 2.5-Pro, and DeepSeek-R1. Each model was provided with a unified prompt that included the source document, the four candidate questions, and instructions to evaluate them on both authenticity and clarity. The evaluation was a forced-choice selection, where the LLM was instructed to identify the single best question for each criterion. The chosen question receives a score of 1, while the others receive 0. The final win rate reported in the main paper is the average score across the three LLM judges. The comprehensive prompt provided to the models is detailed below.

---

**LLM-as-Judge Prompt for Authenticity and Clarity Evaluation**

---

**Instruction:** You are an expert in evaluating the quality of questions related to video games. Please assess the following 4 questions based on two criteria: Authenticity and Clarity.

**Evaluation Criteria:**

- **Authenticity:** Does the question genuinely reflect the actual needs and confusion of game players? Is it based on a real gaming experience?
- **Clarity:** Is the question's expression clear and explicit? Is it easy to understand? Is it specific and not vague?

**Task:** For each criterion, you must choose only one single best question. The chosen question will receive a score of 1, and the others 0. This requires you to carefully compare all questions and select the top performer for that specific dimension.

**Candidate Questions:**

1. [Question 1]
2. [Question 2]
3. [Question 3]
4. [Question 4]

**Output Format:** Please provide your answer strictly in the following JSON format.

```
{
    "authenticity": Chosen question number (integer, e.g., 1),
    "clarity": Chosen question number (integer, e.g., 3),
    "reasoning": {
        "authenticity": "Detailed reasoning for your choice,
        with comparisons to others.",
        "clarity": "Detailed reasoning for your choice, with
        comparisons to others."
    }
}
```

**Notes:** 1. You must select only one question for each criterion (an integer between 1-4). 2. Question numbering starts from 1. 3. You must provide a detailed comparative analysis

---

in your reasoning. 4. Please ensure the JSON format is correct. 5. Return only the JSON object and nothing else.

### H.3 HUMAN EXPERT PROTOCOL

We recruited three domain experts who are veteran players of the target games to provide a ground-truth evaluation. To create the human evaluation set, we randomly sampled 50 instances from each of the three games, for a total of 150 instances and 600 questions.

To ensure a consistent and high-quality annotation process, we provided the experts with a custom-built web interface as shown in Figure 14 and a detailed annotation guideline. For each instance, their majority vote was used as the final decision.

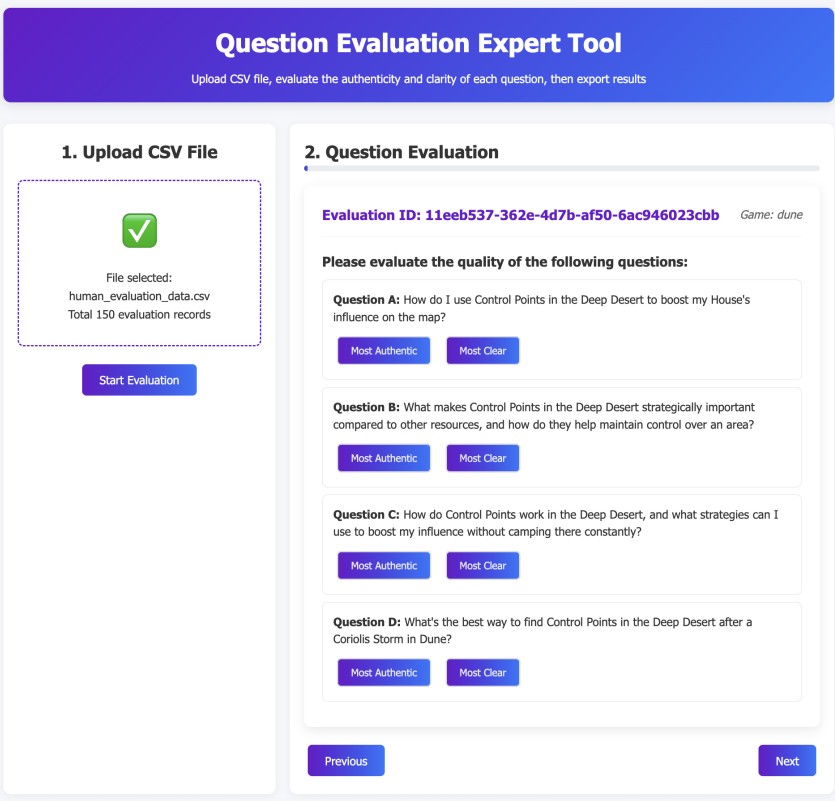

Figure 14: The web-based interface provided to human experts for the 4-way forced-choice evaluation.

### H.4 DETAILED ANALYSIS OF CLARITY EVALUATION

As shown in Figure 15 in the main paper, the evaluation of clarity revealed a notable discrepancy between the judgments of the LLMs and the human experts.

The LLM-as-judges consistently rated questions from the w/o User Persona setting as the clearest. We hypothesize this is because removing the user persona strips the question of its subjective context and conversational phrasing (e.g., a beginner's tone or an expert's implied knowledge). This results in a more direct and objective question. LLMs appear to have a strong bias towards this direct style, perceiving it as less ambiguous and therefore higher in clarity. In contrast, the human experts rated the Full Pipeline as the best for clarity. This suggests that for a domain expert, a clear question is not merely the most direct one. It is a question that effectively uses appropriate personal context to frame a meaningful and unambiguous query. A human might see a question without any persona as contextless and harder to understand its true intent. This finding provides a valuable insight into the behavior of LLM judges: they are highly effective at evaluating objective criteria but may

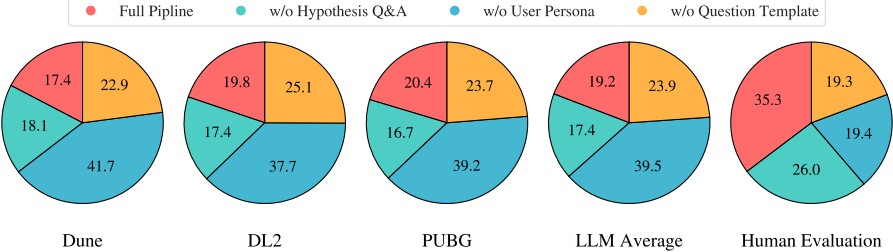

Figure 15: Ablation study results for the clarity criterion. The pie charts show the win rates of our four synthesis settings across the three games, the average score, and the human expert evaluation.

interpret a subjective metric like clarity as structural simplicity. It reinforces the importance of our two-pronged evaluation strategy. By using human experts as the final arbiter for such nuanced criteria, we can confidently validate that our Full Pipeline is superior overall, as it produces questions that are genuinely clear and meaningful to a human audience.

## I GAMERAG-232 LEADERBOARD

In addition to the dynamic lifecycle analysis in the main paper, we constructed a static, high-quality benchmark to provide a comprehensive comparison of various RAG architectures. This allows us to establish a clear leaderboard and analyze the performance of different model combinations in the gaming domain.

### I.1 THE GAMERAG-232 BENCHMARK

To facilitate a broad and rapid comparison of different RAG architectures, we created a high-quality evaluation set, which we term **GameRAG-232**. This dataset consists of 232 expert-verified samples drawn from all three games. Its construction followed a rigorous multi-stage filtering and verification process: (1) We first used our synthesis framework to generate a large pool of 1,000 candidate question-answer pairs for three games. (2) This initial pool was then filtered using embedding-based similarity calculations to remove redundancy, resulting in a set of 428 diverse samples. (3) Finally, these 428 samples were meticulously reviewed by our domain experts, who selected the final 232 high-quality validated samples.

### I.2 GAMERAG-232 LEADERBOARD AND ANALYSIS

We benchmarked 50 distinct end-to-end RAG pipeline combinations on the GameRAG-232 set. To provide a holistic performance metric, we define a **Total Score** as:

$$\text{Total Score} = \text{Correctness} + (\text{Faithfulness}/3)$$

We designed this composite score to place a significantly stronger emphasis on Correctness, as this is the primary indicator of a successful answer for the end-user. In the gaming domain, a factually incorrect answer (low Correctness) is highly detrimental to the user experience, even if it is perfectly faithful to a poorly retrieved document. Faithfulness remains a critical metric, but it is weighted as a secondary component that assesses the system's ability to avoid hallucination. The full leaderboard is presented in Table 8.

ANALYSIS OF LEADERBOARD RESULTS

This new leaderboard provides comprehensive insights into RAG design for the gaming domain:

- **The Critical Role of Reranking:** The most significant finding is the powerful impact of a reranker. The top three ranked pipelines, and seven of the top ten, all utilize the qwen3-reranker-8b. Comparing otherwise identical pipelines (e.g., Rank 2 vs. Rank 8, or Rank 1 vs. Rank 4) shows that the reranker provides a substantial boost in final performance.

- **Retriever Choice is Fundamental:** The choice of the base retriever is critical. Pipelines using a strong dense retriever like text-embedding-3-small vastly outperform their BM25 counterparts.

- **Hybrid Retrieval is not always better** An interesting finding is that hybrid retrieval did not improve performance in our tests. In all comparable cases, the pipelines using only the dense retriever plus a reranker consistently outperformed their hybrid counterparts. This suggests that when a powerful reranker is employed, the additional, less-precise results from BM25 may introduce more noise than signal, slightly hampering the final performance.

- **Generator's Inherent Strengths:** The generator LLM is a key component, as different model families are optimized for distinct capabilities. For instance, the GPT series is noted for its high general correctness. The Gemini series is recognized for its strong faithfulness. These inherent strengths of the generator play a significant role in the quality of the final RAG output.

This new experiment successfully provides a broad comparison of different RAG pipline. We believe this leaderboard is a valuable contribution that provides the community with a clear starting point for selecting RAG pipelines for the gaming domain.

Table 8: RAG Pipeline Leaderboard on the **GameRAG-232** dataset.

| Rank | Retriever(TopK=3) | Reranker | Generator | Correctness | Faithfulness | Total Score |
|------|-------------------|----------|-----------|-------------|--------------|-------------|
| 1 | text-embedding-3-small | qwen3-reranker-8b | o1 | 0.5267 | 0.4353 | 0.6718 |
| 2 | text-embedding-3-small | qwen3-reranker-8b | gemini-2.5-flash | 0.3882 | 0.8362 | 0.6669 |
| 3 | text-embedding-3-small | qwen3-reranker-8b | gemini-2.5-pro | 0.3537 | 0.9095 | 0.6569 |
| 4 | text-embedding-3-small | — | o1 | 0.5000 | 0.4504 | 0.6501 |
| 5 | hybrid (text-emb-3-s + bm25) | qwen3-reranker-8b | o1 | 0.4800 | 0.4950 | 0.6450 |
| 6 | hybrid (text-emb-3-s + bm25) | qwen3-reranker-8b | gemini-2.5-flash | 0.3700 | 0.8040 | 0.6380 |
| 7 | text-embedding-3-small | — | gemini-2.5-pro | 0.3721 | 0.7909 | 0.6357 |
| 8 | text-embedding-3-small | — | gemini-2.5-flash | 0.3685 | 0.7866 | 0.6307 |
| 9 | hybrid (text-emb-3-s + bm25) | qwen3-reranker-8b | gemini-2.5-pro | 0.3400 | 0.8640 | 0.6280 |
| 10 | text-embedding-3-small | qwen3-reranker-8b | o3 | 0.5118 | 0.2909 | 0.6088 |
| 11 | text-embedding-3-small | — | o3 | 0.5129 | 0.2737 | 0.6042 |
| 12 | text-embedding-3-small | qwen3-reranker-8b | gpt-4.1 | 0.4573 | 0.3809 | 0.5843 |
| 13 | text-embedding-3-small | — | gpt-4.1 | 0.4440 | 0.4138 | 0.5819 |
| 14 | text-embedding-3-small | — | gpt-5 | 0.4332 | 0.4397 | 0.5797 |
| 15 | text-embedding-3-small | — | deepseek-v3.2 | 0.3470 | 0.6315 | 0.5575 |
| 16 | bm25 | — | o1 | 0.4052 | 0.4418 | 0.5524 |
| 17 | text-embedding-3-small | qwen3-reranker-8b | gpt-5 | 0.4181 | 0.4030 | 0.5524 |
| 18 | text-embedding-3-small | — | gpt-4o | 0.3793 | 0.5151 | 0.5510 |
| 19 | text-embedding-3-small | qwen3-reranker-8b | gpt-4o | 0.3942 | 0.4698 | 0.5508 |
| 20 | bm25 | — | gpt-4.1 | 0.4181 | 0.3944 | 0.5496 |
| 21 | bm25 | — | gemini-2.5-flash | 0.2974 | 0.7500 | 0.5474 |
| 22 | bm25 | — | gpt-5 | 0.4203 | 0.3793 | 0.5467 |
| 23 | text-embedding-3-small | — | claude-sonnet-4.5 | 0.3405 | 0.5819 | 0.5345 |
| 24 | bm25 | — | gemini-2.5-pro | 0.2866 | 0.7371 | 0.5323 |
| 25 | text-embedding-3-small | — | kimi-k2 | 0.3944 | 0.4095 | 0.5309 |
| 26 | text-embedding-3-small | — | glm-4.5 | 0.3405 | 0.5280 | 0.5165 |
| 27 | text-embedding-3-small | — | gpt-3.5-turbo | 0.3556 | 0.4612 | 0.5093 |
| 28 | bm25 | — | deepseek-v3.2 | 0.2931 | 0.6379 | 0.5057 |
| 29 | text-embedding-3-small | — | claude-3.7-sonnet | 0.3017 | 0.6013 | 0.5022 |
| 30 | bm25 | — | o3 | 0.4095 | 0.2694 | 0.4993 |
| 31 | text-embedding-3-small | — | qwen3-max | 0.3599 | 0.4181 | 0.4993 |
| 32 | bm25 | — | deepseek-v3.1 | 0.2909 | 0.6207 | 0.4978 |
| 33 | text-embedding-3-small | — | claude-sonnet-4 | 0.3103 | 0.5539 | 0.4950 |
| 34 | bm25 | — | gpt-4o | 0.3405 | 0.4483 | 0.4899 |
| 35 | bm25 | — | glm-4.5 | 0.2996 | 0.5431 | 0.4806 |
| 36 | bm25 | — | kimi-k2 | 0.3147 | 0.4569 | 0.4670 |
| 37 | text-embedding-3-small | — | claude-3.5-sonnet | 0.3039 | 0.4806 | 0.4641 |
| 38 | bm25 | — | claude-3.7-sonnet | 0.2522 | 0.6228 | 0.4598 |
| 39 | text-embedding-3-small | — | deepseek-v3.1 | 0.2931 | 0.5000 | 0.4598 |
| 40 | bm25 | — | claude-sonnet-4.5 | 0.2651 | 0.5733 | 0.4562 |
| 41 | bm25 | — | claude-sonnet-4 | 0.2737 | 0.5345 | 0.4519 |
| 42 | bm25 | — | qwen3-max | 0.3082 | 0.4138 | 0.4461 |
| 43 | text-embedding-3-small | — | grok-3 | 0.2888 | 0.4375 | 0.4346 |
| 44 | bm25 | — | gpt-3.5-turbo | 0.2802 | 0.3901 | 0.4102 |
| 45 | bm25 | — | claude-3.5-sonnet | 0.2392 | 0.4978 | 0.4052 |
| 46 | bm25 | — | grok-3 | 0.2150 | 0.3720 | 0.3390 |
| 47 | text-embedding-3-small | — | llama-4-scout-17b | 0.2371 | 0.3707 | 0.3606 |
| 48 | text-embedding-3-small | — | qwen2.5-72b | 0.2694 | 0.2586 | 0.3556 |
| 49 | bm25 | — | qwen2.5-72b | 0.2241 | 0.2478 | 0.3068 |
| 50 | bm25 | — | llama-4-scout-17b | 0.1940 | 0.3362 | 0.3060 |

## J  DE-CONFOUNDING TOPIC SHIFTS FROM INHERENT QUESTION DIFFICULTY

To strengthen our central claims, we conduct a de-confounding analysis to determine whether the observed performance volatility is driven by shifts in topics or by an inherent shift in question difficulty/type. We analyzed the distribution of question types: *Comparative*, *Extraction-based*, *Long-form Answer*, and *Multi-hop Reasoning* across every phase for all three games. The results, as detailed in Tables 9, 10, and 11, show that the distribution of question types remains relatively stable across the entire lifecycle. Crucially, there are no drastic or sudden shifts in the proportion of complex question types that correlate with the significant performance drops observed in our main experiments (e.g., the dip in *DyingLight2* Phase 4). For instance, while Extraction-based QA shows a gradual trend in *DyingLight2* from 62.7% to 73.0%, it lacks any anomalous spike in Phase 4 that could otherwise explain that phase's poor performance.

Table 9: Question type distribution across phases for DUNE.

| Task Type | P1 | P2 | P3 | P4 | P5 | P6 |
|---|---|---|---|---|---|---|
| Comparative | 28.0% | 27.8% | 27.8% | 28.4% | 25.4% | 26.8% |
| Extraction-based | 67.0% | 66.6% | 67.4% | 65.8% | 69.2% | 67.6% |
| Long-form Answer | 4.6% | 4.8% | 4.2% | 5.0% | 4.8% | 4.8% |
| Multi-hop Reasoning | 0.4% | 0.8% | 0.6% | 0.8% | 0.6% | 0.8% |

Table 10: Question type distribution across phases for DyingLight2.

| Task Type | P1 | P2 | P3 | P4 | P5 |
|---|---|---|---|---|---|
| Comparative | 27.0% | 24.8% | 26.2% | 27.8% | 19.8% |
| Extraction-based | 62.7% | 65.2% | 64.0% | 67.8% | 73.0% |
| Long-form Answer | 9.2% | 9.0% | 8.8% | 3.8% | 5.2% |
| Multi-hop Reasoning | 1.0% | 1.0% | 1.0% | 0.8% | 2.0% |

Table 11: Question type distribution across phases for PUBGM.

| Task Type | P1 | P2 | P3 | P4 | P5 | P6 | P7 |
|---|---|---|---|---|---|---|---|
| Comparative | 28.5% | 29.0% | 23.5% | 30.5% | 30.0% | 29.0% | 38.0% |
| Extraction-based | 69.5% | 68.0% | 74.0% | 67.0% | 66.5% | 67.5% | 59.5% |
| Long-form Answer | 1.5% | 2.5% | 2.0% | 2.0% | 3.0% | 2.5% | 2.5% |
| Multi-hop Reasoning | 0.5% | 0.5% | 0.5% | 0.5% | 0.5% | 1.0% | 0.0% |

This stability across phases is an intentional result of our synthesis pipeline's design. As described in Section 3.2.1, our synthesis agent begins by sampling a question type from a predefined set with an initial sampling probability of 25% for all types. This consistent initialization for every phase ensures that the attempt to generate each question type remains uniform across the benchmark's entire lifecycle. The final observed distribution deviates from 25% due to our rigorous Quality Control mechanism, which filters out inauthentic or unanswerable questions. For instance, multi-hop questions are inherently rare in real-world player communities; thus, most synthetic attempts for this type are discarded as being low-quality or inauthentic, ensuring the benchmark remains faithful to actual player behavior while maintaining structural stability.

In conclusion, this analysis allows us to definitively de-confound the experimental variables. The significant performance volatility observed cannot be attributed to a shift in question types, as the distribution remained stable. Instead, the volatility is conclusively driven by topic shifts, proving that specific topics like *Gameplay Mechanics* are inherently more challenging for RAG systems to handle, even for the same basic extraction-based question types. This confirms that *ChronoPlay* successfully captures genuine, topic-driven challenges that would be overlooked by static benchmarks.

## K  EVALUATING KNOWLEDGE-ADAPTIVE RAG SYSTEMS

To further demonstrate the utility of ChronoPlay in evaluating advanced RAG architectures, we conducted an additional experiment comparing a standard append-only baseline with a more sophisticated knowledge-adaptive RAG system. While standard baselines typically handle knowledge updates by simply appending new documents to the corpus, a knowledge-adaptive system actively manages its knowledge base to mitigate the impact of stale information.

The knowledge-adaptive system incorporates a "find-and-replace" update mechanism: (1) Upon the arrival of new knowledge $K_t$, the system extracts key entities to retrieve potentially affected snippets $S_{old}$ from the existing corpus $K_{t-1}$. (2) An LLM agent performs factual validation to identify contradictions between $S_{old}$ and $K_t$. (3) Contradicted snippets are updated or rewritten by the agent to reflect the most recent information before being re-indexed alongside the new knowledge. We evaluated the correctness score of both systems on the Dying Light 2 benchmark using GPT-4o and text-embedding-3-small. The results are presented in Table 12.

Table 12: Comparison between Append-only and Knowledge-adaptive RAG systems on DL2.

| RAG System | P1 | P2 | P3 | P4 | P5 |
|---|---|---|---|---|---|
| Append-only | 0.4225 | 0.4150 | 0.4687 | 0.3887 | 0.3800 |
| Knowledge-adaptive | 0.4225 | 0.4283 | 0.4752 | 0.3921 | 0.4058 |

The experiment yields two primary findings:

- **Quantifying Adaptation Benefits:** In Phase 5, where *Knowledge Evolution* accounts for 42.2% of the environment change (as shown in Figure 8(a)), the knowledge-adaptive system effectively cleaned stale information, improving correctness from 0.3800 to 0.4058. *ChronoPlay* successfully quantifies this gain, proving its ability to measure a system's capacity for factual self-maintenance.

- **The Persistent Challenge of Interest Drift:** Conversely, Phase 4 is almost entirely driven by *User Interest Drift* (31.2% of the change) with minimal knowledge updates. In this phase, the knowledge-adaptive mechanism provided negligible gains (0.3887 vs. 0.3921). The performance volatility remains due to the surge in complex *Gameplay Mechanics* questions, an issue that knowledge-cleaning alone cannot solve.

This de-confounded analysis proves that Dual Dynamics represent two independent challenges. By providing the tools to isolate and measure these forces, ChronoPlay enables researchers to identify whether a RAG system's failure stems from outdated knowledge or an inability to handle shifting user query patterns.

## L  LLM USAGE STATEMENT

In compliance with the ICLR 2026 policy, we disclose the use of a large language model as an assistive tool in the preparation of this manuscript.

The model used was **Gemini 2.5-Pro**. Its role was strictly limited to that of a writing assistant for polishing parts of the text. Specifically, it was used to improve grammar, clarity, and conciseness for author-written content. The LLM was not used for core research ideation, experimental design, data analysis, or the formulation of our conclusions.

Our workflow for using the LLM followed a strict three-step, human-in-the-loop process:

1. **Polish:** We used the model to suggest alternative phrasing or grammatical corrections for existing text drafted by the authors.

2. **Review:** All suggestions provided by the LLM were critically reviewed by the authors to verify their accuracy and to ensure they did not alter the original scientific meaning or intent.

3. **Manual Revision:** We manually integrated and modified any useful suggestions to ensure the final text accurately and precisely reflected our findings and narrative.

The authors take full responsibility for all content presented in this paper, including any text that was revised with the assistance of the LLM.

