# OpenReview forum: "ChronoPlay: A Framework for Modeling Dual Dynamics and Authenticity in Game RAG Benchmarks"
_ICLR.cc/2026/Conference — ICLR 2026 Poster_

### Official Review · Reviewer_2Erh · 2025-10-31

**Soundness:** 2
**Presentation:** 3
**Contribution:** 2
**Rating:** 4
**Confidence:** 4

**Summary:**

This paper focus on the Retrieval Augmented Generation (RAG) problem in dynamic domains and build a RAG benchmark in the domain of online gaming. The authors propose ChronoPlay, a novel framework for the automated and continuous generation of game RAG benchmarks. Specially, this work introduce dual-dynamic update mechanism that responds to changes in both the game’s knowledge and the player’s interests. The experiments across three games demonstrate that RAG system performance is highly volatile over a game’s lifecycle.

**Strengths:**

1. This paper studies the RAG problems in dynamic domains and build a benchmark in this area with an automated and continuous generation framework. The research topic is interesting and valuable.

2. The data sourced from real games and player communities making the benchmark more realistic.

**Weaknesses:**

1. Many real-world applications require dynamic RAG systems, including online shopping (where prices and promotional campaigns constantly change) and travel planning (where weather conditions and seasonal attractions vary). However, the benchmark's coverage of only three games limits its scope and makes the data domain insufficiently diverse.

2. The generation performance in this work is evaluated using LLM-as-Judge, as detailed in Appendix C. However, the meta-evaluation results (Section C.3) are concerning. With accuracy ranging from 70% to 78% and F1-scores from 78% to 86%, the reliability of the evaluation methodology itself is questionable, which undermines confidence in the reported results.

3. Additional experiments on end-to-end RAG systems would provide more comprehensive insights.

**Questions:**

Miss related works on other RAG evaluation frameworks, for example, RAGChecker: A Fine-grained Framework for Diagnosing Retrieval-Augmented Generation by Ru et al., 2024.

---

> ### Author Response · Authors · 2025-11-18
> **Response to W1: On the Scope and Diversity of the Benchmark**
>
> We thank you for highlighting other important dynamic domains like e-commerce and travel. We agree these are valuable areas for future RAG research, and as we note (line 89-line 90), our methodology is indeed applicable to them.
>
> However, as our title "ChronoPlay: A Framework for Modeling Dual Dynamics and Authenticity In **Game** RAG Benchmarks" and abstract  explicitly state, our paper's primary contribution is to provide the first dynamic benchmark methodology for the **gaming domain**. The gaming industry is a vast and highly dynamic digital frontier that, as we established, currently has **no standardized RAG benchmarks**. Our work is focused on filling this critical gap.
>
> We must respectfully but firmly disagree with the assessment that our coverage of only three games limits its scope or is insufficiently diverse. **This scope is not a limitation, but a deliberate and robust demonstration of our framework's generality.** These three games were not chosen at random. They were deliberately selected for their maximum diversity to prove the flexibility of the ChronoPlay framework, as detailed in Appendix A.1. They represent three fundamentally different and challenging dynamic scenarios:
>
> + Dying Light 2 (**Mature Game**): Models gradual, long-term (3.5-year span) knowledge evolution.
> + Dune: Awakening (**New Launch**): Models a volatile launch with massive pre-existing lore and rapid, day-by-day interest shifts ($W=5$ days).
> + PUBG Mobile (**Live-Service**): Models a high-velocity environment with frequent, event-driven updates.
>
> Successfully instantiating our framework across these three fundamentally different types of environments is not a limited scope. It is a robust and extensive demonstration of our framework's generality within our chosen domain, establishing the first benchmark of its kind.

---

> ### Author Response · Authors · 2025-11-18
> **Response to W2: On the Reliability and Validation of the LLM-as-Judge Methodology**
>
> We thank you for this critical look at our evaluation methodology. We want to provide a comprehensive clarification on why our methodology is sound and why the accuracy figures do not undermine our core findings.
>
> First, the use of **LLM-as-Judge has become a standard and widely accepted practice** for large-scale RAG evaluation . Our evaluation are directly inspired by the evaluation protocols established by other major benchmarks, such as CRAG[1].
>
> Second, our paper's primary contribution is not to propose a novel evaluation method. Our core novelty is (1) the ChronoPlay framework for generating dynamic, authentic benchmarks , and (2) using it to discover the Dual Dynamics that cause RAG performance to be highly volatile . The LLM-as-Judge is the tool we use to measure the effects captured by our novel benchmark.
>
> Finally, this 70-78% accuracy does not indicate questionable reliability, but rather a **systematic, conservative bias**. This distinction is key. As shown in Table 4, our judge has **near-perfect 98.77% Precision** on correctness. This means when our judge says an answer is "Good" (Pass), it is almost never wrong. It means our judge is simply **stricter than human experts**. This strictness is a **deliberate and domain-appropriate design choice**. As we argue in the paper (Appendix C.3): **"In a high-stakes domain like gaming, where an incorrect guide or faulty information can directly waste a player’s time and ruin their experience, prioritizing precision over recall is crucial."**  We must use a strict, conservative judge. Thus, this systematic bias may lower the absolute scores, but it does not affect the relative rankings between models or the shape of the fluctuation trends.
>
> Therefore, our judge is a consistent and appropriately conservative tool for measuring relative performance and detecting trends. The methodology is sound.
>
> We certainly acknowledge that other evaluation approaches, such as the more fine-grained diagnostics offered by RAGChecker (which you helpfully pointed out in Q1), are valuable. However, designing a novel, fine-grained evaluation metric is beyond the scope of our paper. Just as the RAGChecker paper focuses its contribution specifically on evaluation, our work focuses its contribution on modeling the specific problem of Dual Dynamics in a generative framework.
>
> [1] Crag-comprehensive rag benchmark. NeurIPS2024.

---

> ### Author Response · Authors · 2025-11-18
> **Response to W3: On the Evaluation of End-to-End RAG Systems**
>
> We thank you for this suggestion to provide more comprehensive insights by testing more end-to-end RAG systems. Our paper already validates the core Dual Dynamics challenge using several of these end-to-end pipelines (e.g., GPT-4o + text-embedding-3). Our experiments on three diverse games and 18 distinct phases (Figure 5 & 6)  already provide strong evidence that this volatility is a real and critical finding.
>
> However, we agree with your point that testing a wider array of these systems is a valuable goal. This aligns perfectly with our other primary contribution: to fill the gap as the first RAG benchmark for the gaming domain.
>
> Therefore, to provide these broader insights, we have conducted a large-scale new experiment, **which we have added as Appendix I in the revised paper**.
>
> **1. A New High-Quality Benchmark GameRAG-232**: First, to facilitate rapid and broad pipeline comparison, we created a high-quality evaluation set GameRAG-232. It consists of 232 expert-verified samples drawn from all three games, following a 3-stage process of generation, de-duplication, and expert validation.
>
> **2. A Comprehensive Comparison for 50 RAG systems**: we benchmarked **50 distinct end-to-end RAG pipelines** on GameRAG-232, using a composite Total Score = Correctness + (Faithfulness / 3) to prioritize factual accuracy. This leaderboard provides exactly the comprehensive insights you requested. For example, our analysis revealed:
>
> + **The Critical Role of Reranking:** The reranker (qwen3-reranker-8b) is important, appearing in 7 of the Top 10 pipelines and providing a substantial boost over non-reranked counterparts.
> + **Retreival Choice is Fundanmental:** The choice of the base retriever is critical. Systems
>   using a strong dense retriever like text-embedding-3-small vastly outperform their BM25
>   counterparts.
>
> + **Hybrid Retrieval is not always better:** When a powerful reranker is used, hybrid retrieval (Dense + BM25) was often worse than using the dense retriever alone, suggesting the BM25 results may add more noise than signal.
> + **Generator's Inherent Strengths:** Different generator families excel at different capabilities. For instance, the GPT series for general correctness, while the Gemini series for high faithfulness.
>
> We believe these new results and the **GameRAG-232** benchmark are a significant contribution, providing the community with a clear starting point for selecting RAG pipelines in the gaming domain.
>
> A sample of this leaderboard (showing the top and bottom results) is below:
>
> | **Rank** | **Retriever**                | **Reranker**      | **Generator**     | **Correctness** | **Faithfulness** | **Total Score** |
> | -------- | ---------------------------- | ----------------- | ----------------- | --------------- | ---------------- | --------------- |
> | 1        | text-embedding-3-small       | qwen3-reranker-8b | o1                | 0.5267          | 0.4353           | 0.6718          |
> | 2        | text-embedding-3-small       | qwen3-reranker-8b | gemini-2.5-flash  | 0.3882          | 0.8362           | 0.6669          |
> | 3        | text-embedding-3-small       | qwen3-reranker-8b | gemini-2.5-pro    | 0.3537          | 0.9095           | 0.6569          |
> | 4        | text-embedding-3-small       | ---               | o1                | 0.5000          | 0.4504           | 0.6501          |
> | 5        | hybrid (text-emb-3-s + bm25) | qwen3-reranker-8b | o1                | 0.4800          | 0.4950           | 0.6450          |
> | 6        | hybrid (text-emb-3-s + bm25) | qwen3-reranker-8b | gemini-2.5-flash  | 0.3700          | 0.8040           | 0.6380          |
> | 7        | text-embedding-3-small       | ---               | gemini-2.5-pro    | 0.3721          | 0.7909           | 0.6357          |
> | 8        | text-embedding-3-small       | ---               | gemini-2.5-flash  | 0.3685          | 0.7866           | 0.6307          |
> | 9        | hybrid (text-emb-3-s + bm25)   | qwen3-reranker-8b | gemini-2.5-pro    | 0.3400          | 0.8640           | 0.6280          |
> | 10       | text-embedding-3-small       | qwen3-reranker-8b | o3                | 0.5118          | 0.2909           | 0.6088          |
> | ...      | ...                          | ...               | ...               | ...             | ...              | ...             |
> | 47       | text-embedding-3-small       | ---               | llama-4-scout-17b | 0.2371          | 0.3707           | 0.3606          |
> | 48       | text-embedding-3-small       | ---               | qwen2.5-72b       | 0.2694          | 0.2586           | 0.3556          |
> | 49       | bm25                         | ---               | qwen2.5-72b       | 0.2241          | 0.2478           | 0.3068          |
> | 50       | bm25                         | ---               | llama-4-scout-17b | 0.1940          | 0.3362           | 0.3060          |

---

> ### Author Response · Authors · 2025-11-18
> **Response to Q1: On Missing Related Work (RAGChecker)**
>
> We thank you for this valuable pointer to RAGChecker. We have reviewed the paper and agree it is an important related work. RAGChecker is a sophisticated evaluation methodology that provides fine-grained, claim-level diagnostics for RAG systems.
>
> This work is complementary to ours, but has a distinct focus. Our ChronoPlay is a benchmark generation framework designed to model a new environmental property: Dual Dynamics. Our original related work (Section 2) was therefore focused on the generation of other **dynamic benchmarks** (e.g., HOH[1], EvolvingQA[2], GrowOVER[3]) , which is where our core novelty lies, rather than on the specific evaluation methods (like RAGChecker) that one runs on those benchmarks.
>
> However, you are correct that RAGChecker is highly relevant to the broader field of RAG evaluation. We agree that adding this discussion will strengthen our related work section, as it will help to clearly distinguish between the contribution of benchmark generation frameworks and evaluation methodologies. We will add a discussion of RAGChecker and other evaluation-centric frameworks to our Related Work section (Section 2) in the final version.
>
> [1] HOH: A Dynamic Benchmark for Evaluating the Impact of Outdated Information on Retrieval-Augmented Generation. ACL2025.
>
> [2] Carpe diem: On the evaluation of world knowledge in lifelong language models. NAACL2024.
>
> [3] Growover: How can llms adapt to growing real-world knowledge? ACL2024.

---

### Official Review · Reviewer_bq56 · 2025-11-01

**Soundness:** 3
**Presentation:** 4
**Contribution:** 3
**Rating:** 6
**Confidence:** 2

**Summary:**

This paper introduces a framework for automatically generating dynamic RAG benchmarks for the gaming domain. The key innovation is addressing "Dual Dynamics"—the simultaneous evolution of game content and player community interests. The framework combines a dual-source synthesis engine that draws from both authoritative game wikis/patch notes and player community discussions to ensure factual correctness and authentic query patterns. The authors instantiate the framework  on three games spanning different timescales and characteristics. Evaluation of various retrieval models (BM25, BGE-M3, Qwen3-Embedding, text-embedding-3) and generator models (GPT-4o, Claude, Gemini, etc.) reveals significant performance fluctuations across game lifecycle phases, demonstrating that both knowledge updates and interest drift independently contribute to benchmark volatility.

**Strengths:**

- The concept of "Dual Dynamics" is a powerful contribution. The paper addresses a limitation of existing dynamic benchmarks and provides a more realistic evaluation paradigm.
- The framework this paper proposes is well-designed.. Human expert evaluation and validation of LLM-as-judge against human experts provide credibility. It is also instantiated on multiple games which shows diversity.

**Weaknesses:**

-  The ChronoPlay pipeline involves multiple LLM-driven stages and several hyperparameters (λ_JSD=0.001, γ=1.5, varying window sizes W). This complexity might pose barriers to easy adoption.
- The statistical rigor of the evaluation could be strengthened. The paper lacks confidence intervals or significance tests for performance differences. Are the fluctuations statistically significant?

**Questions:**

While the authors demonstrate that Dual Dynamics are a major driver of performance volatility, the analysis could be strengthened by de-confounding topic shifts from inherent question difficulty. For instance, analyzing the distribution of question types (e.g., single-hop vs. multi-hop) across phases could provide a more direct measure of difficulty.

---

> ### Author Response · Authors · 2025-11-18
> **Response to W1: On Pipeline Complexity, Hyperparameters, and Ease of Adoption**
>
> We appreciate your attention to our pipeline design. The framework's architecture, including its LLM-driven stages and hyperparameters, is a **purpose-driven design** engineered to solve two specific constraints: **authenticity** and **customizability**.
>
> **1. The LLM-driven stages are necessary for Authenticity:** One of our goal is to achieve **player-centric authenticity**, moving beyond unrealistic synthetic benchmarks. As our ablation study for RQ3 (Figure 7) shows, this goal cannot be met with simpler methods. Removing key modules like the Question Template or User Persona results in a significant loss of authenticity, as judged by both LLMs and human experts . Therefore, these stages are the necessary components required to ensure the benchmark's realism.
>
> **2. The Hyperparameters are features for Customizability:**
>
> Regarding the hyperparameters ($\lambda_{JSD}$, $\gamma$, $W$), these are not barriers, but explicit levers for customization based on domain needs. They allow developers to tune the benchmark's sensitivity.
>
> - For instance, a domain with rapid, high-frequency changes (like the new launch game Dune:Awakening ) can be precisely modeled by setting a very short monitoring window (like $W=5$ days for Dune  to capture fine-grained shifts.
> - Conversely, a mature domain (like DyingLight2) can use a wider window ($W=6$ months) to focus on significant, long-term trends. This adaptability is a core strength.
>
> Finally, to eliminate any barriers to easy adoption, we are fully committed to open-sourcing our work. As demonstrated by the code included in our supplementary materials, we have already taken concrete steps. Upon acceptance, we will publicly release the framework and the instantiated benchmarks to facilitate immediate community adoption.

---

> ### Author Response · Authors · 2025-11-18
> **Response to W2: On Statistical Rigor and the Significance of Performance Fluctuations**
>
> Thank you for this suggestion regarding statistical rigor. We agree that ensuring the observed fluctuations are not artifacts of random noise is critical. We want to clarify why traditional significance tests are not the most central measure of rigor here, due to two key factors:
>
> **1. Our Evaluation is almostly Deterministic:** Significance tests (like p-values) are primarily designed to account for variance introduced by stochasticity (e.g., different training runs). However, our experiment is almostly deterministic:
>
> - In our experimental setup, this source of variance is nearly non-existent. The RAG systems are fixed models.
> - Our LLM-as-Judge evaluation is run almostly deterministically (e.g., with temperature=0.01) to ensure reproducibility. Therefore, re-running our evaluation would produce near identical scores. The variance that significance tests are designed to measure is effectively zero in this context.
>
> **2. Effect Size and Sampling Noise:** another concern is data sampling noise. We address this by focusing on the practical significance, or the sheer magnitude of the fluctuations. As shown in Figure 5c, the correctness for GPT-4o plummets from ~0.43 in Phase 1 to ~0.28 in Phase 2 on the PUBGM benchmark. This is not a marginal 1-2% difference where a p-value would be needed to separate signal from noise. It is a ~34% relative drop in performance.
>
> This massive drop is not a statistical fluctuation but a substantial finding that is directly and causally linked to the Dual Dynamics we model (as proven in Figure 6a). The robustness of our conclusion comes from the deterministic feature of our evaluation combined with the enormous effect size of our findings.

---

> ### Author Response · Authors · 2025-11-18
> **Response to Q1: On De-confounding Topic Shifts from Inherent Question Difficulty**
>
> This is a constructive suggestion. You are correct that to strengthen our claims, we should de-confound whether the performance volatility is driven by Topic Shifts (our hypothesis) or by a shift in inherent question type (your alternative hypothesis).
>
> We analyzed the distribution of question types (Comparative, Extraction-based, Long-form Answer, Multi-hop Reasoning) across every phase for all three games. The results are as follows:
>
> **DUNE**
>
> | Task Type              | Phase_1   | Phase_2   | Phase_3   | Phase_4   | Phase_5   | Phase_6   |
> | ---------------------- | ----------- | ----------- | ----------- | ----------- | ----------- | ----------- |
> | Comparative         | 140 (28.0%) | 139 (27.8%) | 139 (27.8%) | 142 (28.4%) | 127 (25.4%) | 134 (26.8%) |
> | Extraction-based    | 335 (67.0%) | 333 (66.6%) | 337 (67.4%) | 329 (65.8%) | 346 (69.2%) | 338 (67.6%) |
> | Long-form Answer | 23 (4.6%)   | 24 (4.8%)   | 21 (4.2%)   | 25 (5.0%)   | 24 (4.8%)   | 24 (4.8%)   |
> | Multi-hop Reasoning | 2 (0.4%)    | 4 (0.8%)    | 3 (0.6%)    | 4 (0.8%)    | 3 (0.6%)    | 4 (0.8%)    |
>
> **DyingLight2**
>
> | Task Type              | Phase_1   | Phase_2   | Phase_3   | Phase_4   | Phase_5   |
> | ---------------------- | ----------- | ----------- | ----------- | ----------- | ----------- |
> | Comparative         | 108 (27.0%) | 99 (24.8%)  | 105 (26.2%) | 111 (27.8%) | 79 (19.8%)  |
> | Extraction-based    | 251 (62.7%) | 261 (65.2%) | 256 (64.0%) | 271 (67.8%) | 292 (73.0%) |
> | Long-form Answer    | 37 (9.2%)   | 36 (9.0%)   | 35 (8.8%)   | 15 (3.8%)   | 21 (5.2%)   |
> | Multi-hop Reasoning | 4 (1.0%)    | 4 (1.0%)    | 4 (1.0%)    | 3 (0.8%)    | 8 (2.0%)    |
>
> **PUBGM**
>
> | Task Type              | Phase_1   | Phase_2   | Phase_3   | Phase_4   | Phase_5   | Phase_6   | Phase_7   |
> | ---------------------- | ----------- | ----------- | ----------- | ----------- | ----------- | ----------- | ----------- |
> | Comparative| 57 (28.5%)  | 58 (29.0%)  | 47 (23.5%)  | 61 (30.5%)  | 60 (30.0%)  | 58 (29.0%)  | 76 (38.0%)  |
> | Extraction-based    | 139 (69.5%) | 136 (68.0%) | 148 (74.0%) | 134 (67.0%) | 133 (66.5%) | 135 (67.5%) | 119 (59.5%) |
> | Long-form Answer    | 3 (1.5%)    | 5 (2.5%)    | 4 (2.0%)    | 4 (2.0%)    | 6 (3.0%)    | 5 (2.5%)    | 5 (2.5%)    |
> | Multi-hop Reasoning | 1 (0.5%)    | 1 (0.5%)    | 1 (0.5%)    | 1 (0.5%)    | 1 (0.5%)    | 2 (1.0%)    | 0 (0.0%)    |
>
> **1. Analysis:** The results show a crucial finding: the distribution of question types is **relatively stable** across all phases. We mean there are **no drastic, sudden shifts** in question type that would correlate with and explain the major performance drops we observe (e.g., in DyingLight2 Phase 4). For instance, in DyingLight2, Extraction-based QA (the most common type) shows a gradual trend (from 62.7% in P1 to 73.0% in Phase5), but it does not show a sudden, anomalous spike specifically in Phase 4 (67.8%) that would explain that phase's poor performance. We observed this same pattern in the DUNE and PUBGM benchmarks as well.
>
> **2. Why this Stability Exists:** This stability across phases is an intentional and crucial result of our synthesis pipeline's design.
>
> As described in Section 3.2.1, our synthesis agent begins by sampling a question type $q_t$ from a predefined set. We set the **initial sampling probability for all types to an equal 25%**. This identical setting, used for all phases, ensures that our attempt to generate each question type is perfectly consistent across the benchmark's entire lifecycle. This is why the phase-to-phase distribution remains stable.
>
> You correctly noted that the final distribution is not 25%. This is not a flaw, but rather the intended and successful result of our quality control mechanism. This synthesis agent filters out unrealistic or unanswerable questions. Multi-hop questions are rare in real player communities. Therefore, most of our synthetic Multi-hop attempts are discarded during the quality control loop as being low-quality or inauthentic.
>
> **3. Conclusion:** This new analysis allows us to definitively de-confound the variables. The significant performance volatility observed in our experiments cannot be attributed to a shift in question type, because that distribution remained stable. This proves that the volatility is conclusively driven by the topic shift. This strengthens our original analysis (Sec 4.2.1 & Appendix G)  and makes it more precise: the topic of GAMEPLAY_MECHANICS is inherently more difficult for RAG systems to handle, even for the same simple extraction-based question types.
>
> This confirms that ChronoPlay successfully captures a genuine, topic-driven challenge that static benchmarks would miss. We sincerely thank you for this suggestion, as it has made our paper's central claim significantly stronger. We will add this new analysis to the Appendix.

---

> ### Comment · Reviewer_bq56 · 2025-11-23
>
> I appreciate the authors' efforts in the additional analysis. I have raised the soundness score. Since the rebuttal has partially addressed my concerns, and I am happy to maintain my original overall positive assessment.

---

> > ### Author Response · Authors · 2025-11-25
> >
> > Thank you for taking the time to review our responses and the additional analysis. Your insightful feedback has been invaluable in helping us clarify and improve our work, and we will incorporate all relevant content in the final version. Lastly, we sincerely appreciate your continued positive assessment and are especially grateful for the improved soundness score.

---

### Official Review · Reviewer_YZgc · 2025-11-05

**Soundness:** 3
**Presentation:** 4
**Contribution:** 3
**Rating:** 6
**Confidence:** 4

**Summary:**

This paper introduces ChronoPlay, a framework for constructing dynamic retrieval-augmented generation (RAG) benchmarks in gaming domains. It models two evolving dimensions: knowledge evolution (continuous updates to game rules and content) and user interest drift (shifts in player community focus over time). The framework integrates an authoritative knowledge base, community-derived question templates, and synthetic player personas to generate realistic, time-evolving question–answer pairs.

The authors test various retrievers and generators on datasets created with ChronoPlay for three games — Dying Light 2, Dune: Awakening, and PUBG Mobile — and provide detailed ablation studies.

**Strengths:**

1. ChronoPlay is the first framework to incorporate dynamic, evolving environments into RAG evaluation for gaming. By formalizing knowledge evolution and user interest drift, it highlights two important factors often overlooked in static RAG benchmarks.

2. The experiments are extensive, covering several retrievers and generators across three games and including ablations on knowledge and interest dynamics.

3. The proposed datasets and methodology could help the community study how RAG systems behave under temporal and contextual changes.

**Weaknesses:**

1. While the paper includes a diverse set of retrievers and generators, the RAG experiments themselves are static. It's unclear whether the RAG system re-index as the benchmark evolves. There is no adaptive RAG system (e.g. with finetuning, memory) discussed. It's unclear how the proposed dynamic benchmark would challenge or benefit adaptive RAG systems in practice.

2. The paper also omits details about how the vector databases are indexed or re-indexed after each phase. It is unclear whether embeddings are refreshed, incrementally updated, or kept static.

3. ChronoPlay models a population-level community through generated player personas, but it doesn’t capture personalized or history-dependent questions. I don’t see this as a major weakness, but it does make me wonder how this limitation relates to the idea of dynamic RAG. What would a RAG system need to do to account for personalized shifts in interest over time?

**Questions:**

1. How are vector databases indexed or re-indexed after each phase update?
2. Could ChronoPlay be extended to simulate user-specific histories or personalized question evolution, beyond aggregate community changes?

---

> ### Author Response · Authors · 2025-11-18
> **Response to W1, W2, & Q1: Clarification on Re-indexing and the Evaluation of Adaptive RAG Systems**
>
> We thank you for this excellent question. We will first clarify our baseline's re-indexing process (W2, Q1) and then present a new experiment (inspired by W1) that directly demonstrates how ChronoPlay benefits the evaluation of knowledge-adaptive RAG systems.
>
>
>
> **1. On Re-indexing in Our Original Baselines (W2 & Q1):** Our baseline setup (Section 4.2) models a **static re-index** approach. As defined in Section 3.3 1, the knowledge base is updated via $\mathcal{K}\_{t+1}=\mathcal{K}\_{t} \cup \mathcal{A}\_{new}$. This means new knowledge $\mathcal{A}\_{new}$ is added to the existing corpus $\mathcal{K}\_{t}$. The RAG baselines then **re-index the entire growing corpus $\mathcal{K}_{t+1}$** from scratch for each phase.
>
> **2. New Experiment: A Knowledge-Adaptive RAG System (W1):** Inspired by your suggestions, we developed a new **knowledge-adaptive RAG system** to test on our benchmark. This system actively maintains its knowledge base:
>
> + When new knowledge $\mathcal{A}\_{new}$ arrives, we first use its extracted entities $\sigma\_{update}$  to retrieve a subset of potentially affected snippets $k_i$ from the old corpus $\mathcal{K}\_{t}$.
> + Then, an LLM agent performs a "find-and-replace" operation: it validates if $k_i$ is factually contradicted by $\mathcal{A}_{new}$ and **updates (re-writes) $k_i$ with the new information.**
> + The new knowledge $\mathcal{A}_{new}$ is also added as a new snippet.
> + This results in a cleaner, more accurate knowledge base $\mathcal{K}_{t+1}$ which is then re-indexed. (We note this process is computationally intensive, as it involves LLM-based validation of many retrieved snippets).
>
> We evaluated this new knowledge-adaptive system (using text-embedding-3 +GPT-4o) against our original append-only baseline on the Dying Light 2 benchmark.The results (Correctness Scores) are as follows:
>
> | **RAG System**                  | **Phase 1** | **Phase 2** | **Phase 3** | **Phase 4** | **Phase 5** |
> | ------------------------------- | ----------- | ----------- | ----------- | ----------- | ----------- |
> | append-only system (Baseline)   | 0.4225      | 0.4150      | 0.4687      | 0.3887      | 0.3800      |
> | knowledge-adaptive system (New) | 0.4225      | 0.4283      | 0.4752      | 0.3921      | 0.4058      |
>
> **Finding 1 (Benefit):** As shown in Figure 8(a) of our paper, the update to Phase 5 was heavily driven by **Knowledge Evolution (42.2% of the change)**. In this phase, our new knowledge-adaptive system, by successfully cleaning stale information, improved its Correctness score from 0.3800 to 0.4058. ChronoPlay is able to precisely quantify the benefit of this adaptation.
>
> **Finding 2 (The New Challenge):** Conversely, Figure 8(a) shows that the update to Phase 4 was almost entirely driven by user interest drift (31.2% of the change), with no knowledge updates. As our new experiment shows, the knowledge-adaptive system provided minor performance gain in this phase (0.3887 vs. 0.3921). The performance volatility caused by the topic shift remained, which we identified in Sec 4.2.1 as a surge in complex GAMEPLAY_MECHANICS questions.
>
> (**Note:** The append-only baseline still have opportunities to correctly answer questions about updates if its retriever happens to retrieve the new snippet)
>
> This proves our central motivation: Dual Dynamics are two independent challenges. ChronoPlay provides the precise challenge and measurement tools needed to test both forms of adaptation, which is a significant benefit to any researcher building truly adaptive RAGs. We will add this new experiment to the final paper.

---

> ### Author Response · Authors · 2025-11-18
> **Response to W3 & Q2: On Personalized Histories vs. Aggregate-Level Dynamics**
>
> Thank you for these insightful questions. You are correctly pointing out two distinct but related frontiers for dynamic RAG: the design of the **benchmark (Q2)** and the design of the **RAG system (W3)**. We will address the benchmark question first, as it sets the context for the system design.
>
> **1. On Extending the Benchmark (Q2):** Our current framework is designed to model dynamics at the population level. This was a conscious choice to ensure our benchmark is general and robust. This population-level approach is essential for building a public benchmark that captures the overall state of the player community.
>
> To your question (Q2) about extending this to user-specific histories, this is an interesting but non-trivial challenge. As you noted, this would require tracking an individual user's question history over time. Based on our experience collecting data from public community, this presents several significant challenges:
>
> 1. **Tracking & Anonymity:** Reliably tracking a single user across multiple platforms (e.g., Reddit, Fandom, Discord) and over a long period is technically difficult because of cross-platform anonymity and raises data privacy questions.
> 2. **Data Scarcity & Sparsity:** Even if an individual user can be tracked, their question history is often sparse. It would be difficult to build a robust, large-scale evaluation set from the high-quality questions posted by a single person.
>
> However, if these challenges were overcome (e.g., in a private data setting where user histories are available), the ChronoPlay framework is ideally suited for this extension.
>
> The core modification would be to our User Persona Base $\mathcal{U}_{comm}$.
>
> + **Current (Aggregate):** The base is a static collection of aggregate persona types, and our synthesis pipeline samples from it.
>
> - **Future (Personalized):** This module could be replaced with a **dynamic user state module**. This new module would take a specific user_id as input and maintain a stateful profile of that user's history and progress.
>
> The rest of the ChronoPlay pipeline would then generate questions conditioned on this dynamic user state rather than a aggregate persona.
>
> **2. On Designing the RAG System (W3):** Given the challenges in benchmarking personalization, we can now address your second question: "What would a RAG system need to do to account for personalized shifts in interest over time?"
>
> We think a RAG system would require at least three components to handle this personalized dynamic:
>
> 1. **A Stateful User Model:** The system must move beyond stateless queries and maintain a persistent profile or state for each user.
> 2. **A State Update Mechanism:** This user state must be dynamically updated based on their interactions.
> 3. **A State-Conditioned RAG Pipeline:** The user's state must influence the RAG pipeline in real-time. This could happen in two ways:
>    - **Personalized Retrieval:** The user's state could be used to augment the query, guiding the retriever towards beginner guides rather than end-game wikis.
>    - **Personalized Generation:** The generator's prompt could be conditioned on this state. This is analogous to how our framework uses the User Persona Base, but here it would be dynamic and individual, allowing the LLM to provide answers tailored to the user's specific context.
>
> Therefore, our aggregate-level modeling is the robust and principled choice for a general benchmark. We agree that personalized dynamic RAG is a key frontier, and we will add this discussion of the opportunities to our Conclusion and Future Work section.

---

> > ### Comment · Reviewer_YZgc · 2025-11-26
> >
> > I appreciate the authors’ response and the new experiments, which addressed my concerns well. I maintain my original recommendation to accept this paper and hope the other reviewers will be convinced as well.

---

> > > ### Author Response · Authors · 2025-11-27
> > >
> > > We sincerely appreciate your positive feedback and your continued support for our work. We are glad to hear that our response and the new experiments have satisfactorily addressed your concerns. We will ensure that these new experimental results and the corresponding discussions are incorporated into the final version of the paper.

---

### Author Response · Authors · 2025-11-29
**Summary of Rebuttal Phase & Reviewer Consensus**

Dear Program Chairs, Senior Area Chairs, and Area Chair,

We provide this summary of the rebuttal phase to highlight the constructive dialogue we had with the reviewers and the **substantial improvements** made to the paper based on their insightful suggestions.

**1. Consensus on Strengths**

We are encouraged that the novelty and core contribution of our work were recognized by all reviewers at the outset:

- *Novelty of Dual Dynamics*: **Reviewer YZgc** recognized ChronoPlay as the **first** framework to incorporate dynamic, evolving environments into gaming RAG evaluation. **Reviewer bq56** highlighted the concept of "Dual Dynamics" as a **powerful contribution** that addresses limitations of existing static benchmarks. **Reviewer 2Erh** found the research topic **interesting and valuable**.
- *Realistic & Automated Framework:* **Reviewer bq56** stated the framework is **well-designed**. **Reviewer 2Erh** praised the automated and continuous generation framework and noted that sourcing data from real games and player communities makes the benchmark **more realistic**. **Reviewer YZgc** noted it captures factors **often overlooked in static RAG benchmarks**.
- *Extensive Experimentation:* **Reviewer YZgc** commended the extensive experiments covering multiple retrievers/generators across three games. **Reviewer bq56** highlighted that the instantiation on multiple games **shows diversity**.

**2. Productive Dialogue with Reviewers YZgc and bq56**

We actively addressed the concerns raised by Reviewers YZgc and bq56, leading to a positive consensus:

- **Reviewer YZgc** raised questions regarding the evaluation of **adaptive RAG systems** (W1), the **re-indexing process** (W2,Q1), and the potential for **personalized histories** (W3,Q2).
  - We clarified the re-indexing baseline and implemented a new **Knowledge-Adaptive RAG system**. The results proved that ChronoPlay can precisely quantify the benefits of knowledge updates while highlighting the independent challenge of topic drift. We also provided a detailed discussion on the roadmap from aggregate to personalized dynamics.
  - **Reviewer YZgc** stated the new experiments **"addressed my concerns well"** and explicitly maintained their recommendation to **accept**.
- **Reviewer bq56** expressed concerns about **pipeline complexity** (W1), **statistical rigor** (W2) and the need to **de-confound topic shifts from question difficulty** (Q1).
  - We justified the pipeline complexity as essential for authenticity and explained that the deterministic evaluation with massive effect sizes ensures statistical robustness. Additionally, our statistical analysis of question types confirmed that volatility is driven by topic shifts rather than difficulty spikes.
  - **Reviewer bq56** appreciated the additional analysis and stated **"happy to maintain my original overall positive assessment"** and explicitly **raised the Soundness score**.

**3. Comprehensive Efforts to Address Reviewer 2Erh**

While Reviewer 2Erh did not have the opportunity to respond due to the system incident, we have incorporated major updates to fully resolve his concerns:

- **On End-to-End RAG Systems (W3):**  Reviewer 2Erh requested more diverse end-to-end RAG system evaluations.

  We conducted a massive new experiment (added in **Appendix I**). We constructed a high-quality evaluation set **GameRAG-232** and benchmarked **50 distinct RAG pipelines**. This new experiment provides the comprehensive insights requested and establishes a robust baseline for the community.

- **On Scope (W1):**  Reviewer 2Erh questioned if covering only three games limited the diversity.

  We clarified that the three games were deliberately selected to represent three fundamentally different dynamic scenarios: *Mature* (Dying Light 2), *New Launch* (Dune), and *Live-Service* (PUBGM), ensuring the framework is robust across different timescales.

- **On Evaluation (W2):**  Reviewer 2Erh raised concerns about the LLM-judge accuracy.

  We clarified that in the gaming domain, incorrect guides or faulty information can directly **waste a player’s time and ruin their experience**. Therefore, our judge is designed to prioritize precision (98.77%) over accuracy. This conservative design ensures that the benchmark does not reward hallucinated advice and that the relative rankings between models remain reliable.

- **On Related Work (Q1):**  Reviewer 2Erh suggested discussing RAGChecker.

  We agreed that RAGChecker is a valuable complementary work. We will update the Related Work section to discuss it, effectively using it to clarify the **distinction between our benchmark generation framework and evaluation methodologies**.

We trust that these comprehensive additional experiments and detailed clarifications satisfactorily resolve the concerns raised by Reviewer 2Erh.

---

### Meta-Review · Area_Chair_GmHA · 2026-01-07

**Summary:**

There are mainly seven scattered concerns raised by the three reviewers. Reviewer YZgc notes that the paper does not discuss adaptive RAG systems, such as those incorporating finetuning or memory mechanisms, and also finds it unclear whether the embeddings are refreshed, incrementally updated, or kept static. Reviewer bq56 expresses concerns about the involvement of multiple LLM-driven stages together with several hyperparameters, and suggests that the statistical rigor of the evaluation could be further strengthened. Reviewer 2Erh points out that the benchmark covers only three games, which limits its scope and results in insufficient data-domain diversity, questions the reliability of the evaluation methodology itself, and requests additional experiments on end-to-end RAG systems.

**Reviewer Concerns:**

Among the reviewers’ concerns, several were addressed by the rebuttal through explanation and clarification. The authors clarified that the work already considers a knowledge-adaptive RAG setting, resolving the concern about the absence of an adaptive system, and explicitly stated that the baseline uses a static re-indexing approach, addressing the ambiguity around embedding updates. The involvement of multiple LLM-driven stages and hyperparameters was justified by positioning aggregate-level modeling as a robust and principled choice for a general-purpose benchmark. Concerns about evaluation reliability were addressed by noting that LLM-as-Judge is now a widely accepted standard and that the paper does not aim to propose a novel evaluation method. A large-scale end-to-end RAG experiment was also added in Appendix I. Remaining concerns include the statistical rigor of the evaluation, where deprioritizing traditional significance tests has not been fully endorsed, and the benchmark’s limited coverage of only three games, which does not completely address the reviewer’s concern about domain diversity.

**Reviewer Scores:**

Reviewer YZgc and Reviewer bq56 have their concerns addressed or partially addressed, and both are expected to maintain their current scores of six. For Reviewer 2Erh, some concerns have been addressed, but the remaining issue regarding the benchmark’s limited coverage means the reviewer might either increase the score to six or keep the current score of four.

---

### Decision · Program_Chairs · 2026-01-26

Accept (Poster)